# EqR: Equivariant Representations for Data-Efficient Reinforcement Learning

## Abstract

We study different notions of equivariance as an inductive bias in Reinforcement Learning (RL) and propose new mechanisms for recovering representations that are equivariant to both an agent's action, and symmetry transformations of the state-action pairs. Whereas prior work on exploiting symmetries in deep RL can only incorporate predefined linear transformations, our approach allows for non-linear symmetry transformations of state-action pairs to be learned from the data itself. This is achieved through an equivariant Lie algebraic parameterization of state and action encodings, equivariant latent transition models, and the use of symmetry-based losses. We demonstrate the advantages of our learned equivariant representations for Atari games, in a data-efficient setting limited to 100K steps of interactions with the environment. Our method, which we call Equivariant representations for RL (EqR), outperforms other comparable methods on statistically reliable evaluation metrics.

## 1 Introduction

The recent success of deep reinforcement learning (François-Lavet et al., 2018) in applications to games such as Atari (Mnih et al., 2015), Go (Silver et al., 2016) and Poker (Brown & Sandholm, 2019), to applications in robotics (Levine et al., 2016) and autonomous navigation (Bellemare et al., 2020) has demonstrated its promise as the framework of choice for sequential decision making. However, the use of a reward as the only signal for representation learning with high dimensional states and actions leads to tremendous data inefficiency. Notably, almost all success stories of RL rely on vast amounts of data or simulations with a huge computational overload.

More data-efficient representation learning (Bengio et al., 2013) requires stronger inductive biases, though the search for a general yet strong inductive bias is still under way. One general approach is to place a central role on *transformations* of the data, where invariance and equivariance to a set of transformations imposes strong conditions on the learned representations. This viewpoint is particularly appealing in RL, where the agent is in control of some of these transformations through its own actions. Fig. 1 illustrates this concept using the example of a 2D pendulum. Moreover,

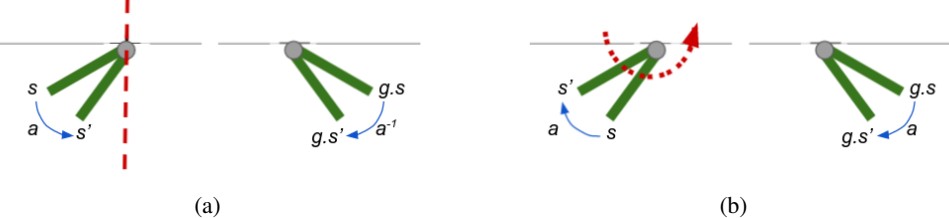

(a)            (b)

Figure 1: An illustration of typical symmetries in a pendulum, and the corresponding transformations of the state and action for a group equivariant transition model: (a) shows how reflection of the agent's state results in a permutation of the action, denoted by $a^{-1}$. (b) shows how rotation of the agent's state results in invariance of the action in the absence of gravity. The state transitions can be modeled as group actions (2D rotations in this example), which can be captured by our symmetry transformation-based transition model. Note that rotational symmetry can hold even when gravity is present. In this case, symmetry transformations include rotations (and reflections) that preserve the Hamiltonian. Such non-linear energy-preserving transformations of state-actions in the pixel space can become linear in the embedding space.

transformations naturally lead to a notion of disentanglement in the representations (Higgins et al., 2018), potentially enabling better out-of-distribution generalization (Higgins et al., 2017; Thomas et al., 2017). The recent success of self-supervised learning approaches that rely on a (predefined) set of transformations (Chen et al., 2020; Zbontar et al., 2021), and also within the context of RL (Yarats et al., 2021; Laskin et al., 2020b), further highlights the importance of transformations in data-efficient representation learning.

Motivated by these observations, this work develops a broader perspective on the notion of equivariant representation learning within RL. In particular, we integrate equivariance under the agent's action and equivariance under the symmetries of the environment into a single latent variable model that is equivariant to an *a priori unknown* group of non-linear transformations of state-action pairs. In contrast to the traditional approach of using symmetric Markov Decision Processes (MDPs), we argue for modeling the larger group of state-action symmetries (separate from reward symmetries), and show how to parameterize the latent embeddings of states and actions to make the representations equivariant to continuous transformations of the environment resulting from agent's action. We benchmark our approach, which we call Equivariant representations for RL (EqR), on the 26 games in the Atari 100K benchmark (Kaiser et al., 2019), where we outperform other comparable methods using reliable evaluation metrics (Agarwal et al., 2021). Our approach, however, is not restricted to this domain. It is applicable in any setting where the transformations that an agent undergoes can be expressed using matrix Lie groups, including autonomous driving, navigation, and robotics.

## 2 RELATED WORK

The use of transformations, be it in data-augmentation or self-supervision, has become a common ingredient in recent representation learning methods for deep RL. However, theoretical work on symmetry in RL goes back to Zinkevich & Balch (2001) and Ravindran & Barto (2001), both of which use symmetric MDPs. A more recent use of this formalism is in van der Pol et al. (2020b); Mondal et al. (2020), where policy networks, with built-in equivariance, are shown to improve data-efficiency. Closely related notions, that motivated the early work on symmetric MDPs, are model minimization (Ravindran & Barto, 2002), state abstraction (Ravindran & Barto, 2003; Li et al., 2006), MDP homomorphism (Ravindran & Barto, 2004) and lax bisimulations (Taylor, 2008). In particular, MDP homomorphism, which requires equivariance under an agent's action, encompasses the general idea of model-based reinforcement learning. As examples, a latent MDP that matches the state dynamics and the reward distribution of the environment is learned in (van der Pol et al., 2020a; Gelada et al., 2019).

Other works in RL that are relevant to our objective are those that attempt to increase data-efficiency using a learned model of the environment. While some methods such as SimPLe (Kaiser et al., 2019), learn this transition model at the pixel level, the majority of methods use a latent space model. The latent space is either learned using reconstruction (Hafner et al., 2019a;b), or through self-supervision and contrastive methods (CURL, Laskin et al., 2020b). However, there is evidence that the improvement in sample efficiency is largely due to image augmentation, as seen in Laskin et al. (2020a) and DrQ (Yarats et al., 2021). Using a reconstruction-based method is also inefficient because similar to pixel level models, one needs to learn potentially irrelevant details. The fact that variations of model-free algorithms such as Data-Efficient Rainbow (DER) (van Hasselt et al., 2019) and OTRainbow (Kielak, 2019) can achieve a similar performance to reconstruction-based methods without explicit representation learning components confirms this intuition. More recently SPR (Schwarzer et al., 2021) shows that data augmentation and improvements in Rainbow combined with particular forms of self-supervision can significantly improve the sample efficiency, producing state-of-the-art results in sample-efficient representation learning in RL.

## 3 BACKGROUND

### 3.1 GROUPS AND THEIR REPRESENTATIONS

A group $\mathcal{G} = \{g\}$ is a set, equipped with an associative binary operation, such that the set is closed under this operation, and each element $g \in \mathcal{G}$ has a unique inverse, such that their composition gives the identity $g^{-1}g = e$. Any subset $\mathcal{G}' \leq \mathcal{G}$ that is closed under binary operation of the groups forms a subgroup. A group $\mathcal{G}$ can *act* on a set $\mathbb{X}$ by transforming its elements $x \in \mathbb{X}$ through a

bijection. We use $\alpha : \mathcal{G} \times \mathbb{X} \mapsto \mathbb{X}$ to denote the *group action*, and for brevity replace $\alpha(g, x)$ with $g \cdot x$ moving forward. The action captures some of the structure of $\mathcal{G}$ due to two constraints – the identity element acts trivially $e \cdot x = x$; and composition of actions is equal to action of the composition, i.e., $(gg') \cdot x = g \cdot (g' \cdot x), \forall g, g' \in \mathcal{G}$. $\mathbb{X}$ is then called a $\mathcal{G}$-set. Any $\mathcal{G}$-action partitions $\mathbb{X}$ into *orbits* $x^{\mathcal{G}} = \{g \cdot x \mid g \in \mathcal{G}\}$, and we denote the set of orbits under $\mathcal{G}$-action using $\mathbb{X}/\mathcal{G}$. A $\mathcal{G}$-action is transitive iff its action results in a single orbit.

**Parameterizing Lie Groups** In this work, we assume $\mathcal{G}$ is (any sub-group of) a *classical Lie group* over $\mathbb{R}$. These are the groups that can be represented using invertible matrices – in other words, we consider groups whose action is linear on real vector spaces. We use $\rho(\mathcal{G})$ to denote a linear representation of $\mathcal{G}$, and $\rho_g : \mathbb{R}^D \to \mathbb{R}^D$ for the action (a.k.a. the representation) of $g \in \mathcal{G}$. Two other greek letters $\tau$ and $\kappa$ are also used for this purpose. Many such Lie groups are identifiable by their *Lie algebra* $\mathfrak{g} = Lie(\rho(\mathcal{G}))$.[1] This connection enables a simple parameterization of $\rho(\mathcal{G})$ using a set of linear bases for their Lie algebra – that is $\rho_g = \exp(\sum_i \beta_{g,i} \mathbf{E}^{(i)})$, where $\exp(\mathbf{Y}) = \sum_{j=0}^{\infty} \frac{\mathbf{Y}^j}{j!}$ is the matrix exponential. We refer to this parameterization later in Section 4. Such linear representations in the form of invertible matrices can be used for both continuous transformations (*e.g.*, 3D rotation) and finite groups ($\times 90°$ rotations).

## 3.2 MDP Homomorphism and Symmetric MDPs

We define an MDP as the 4-tuple $\mathcal{M} = \langle \mathbb{S}, \mathbb{A}, R, T \rangle$ where $\mathbb{S}$ and $\mathbb{A}$ are respectively the set of states and actions, $R : \mathbb{S} \times \mathbb{A} \to \mathbb{R}$ is the reward function, and $T : \mathbb{S} \times \mathbb{A} \times \mathbb{S} \to \mathbb{R}^{\geq 0}$ is the state transition function.[2] For two MDPs $\mathcal{M} = \langle \mathbb{S}, \mathbb{A}, R, T \rangle$ and $\bar{\mathcal{M}} = \langle \bar{\mathbb{S}}, \bar{\mathbb{A}}, \bar{R}, \bar{T} \rangle$, MDP homomorphism can be defined as a tuple $\mathcal{H} = \langle h_{\mathbb{S}}, h_{\mathbb{A}} \rangle$ where $h_{\mathbb{S}} : \mathbb{S} \to \bar{\mathbb{S}}$ is the state mapping and $h_{\mathbb{A}} : \mathbb{S} \times \mathbb{A} \to \bar{\mathbb{A}}$ is the state dependent action mapping. These two mappings satisfy the following invariance and equivariance conditions:

1) Invariance of the reward:

$$\bar{R}(h_{\mathbb{S}}(s), h_{\mathbb{A}}(s, a)) = R(s, a) \quad \forall s, a \in \mathbb{S} \times \mathbb{A} \tag{1}$$

2) Equivariance of the deterministic transition model under the agent's action:

$$\bar{T}(h_{\mathbb{S}}(s), h_{\mathbb{A}}(s, a)) = h_{\mathbb{S}}(T(s, a)) \quad \forall s, a \in \mathbb{S} \times \mathbb{A} \tag{2}$$

A probabilistic variation of the above equation for a stochastic MDP (Bloem-Reddy & Teh, 2020) is:

$$\bar{T}(h_{\mathbb{S}}(s') \mid h_{\mathbb{S}}(s), h_{\mathbb{A}}(s, a)) = \sum_{s'' \in [s']_h} T(s'' \mid s, a) \quad \forall s, s' \in \mathbb{S}, a \in \mathbb{A}, \tag{3}$$

where $[s']_{h_{\mathbb{S}}} = h_{\mathbb{S}}^{-1}(h_{\mathbb{S}}(s'))$ is the equivalence class of $s'$ under $h_{\mathbb{S}}$.

In related literature, MDP homomorphism is often used for minimization of the MDP, because the optimal policy of $\bar{\mathcal{M}}$ can be lifted to obtain the optimal counterparts for $\mathcal{M}$.

**Symmetric MDPs** The automorphism group $\mathcal{G}_{\mathcal{M}} = \text{Aut}(\mathcal{M})$ of an MDP identifies the set of symmetry transformations of state-actions that preserve the reward and the transition dynamics:

$$R(s, a) = R(g \cdot \langle s, a \rangle)) \qquad \forall g \in \mathcal{G}_{\mathcal{M}}, s \in \mathbb{S}, a \in \mathbb{A} \tag{4}$$

$$T(s' \mid s, a) = T(g \cdot s' \mid g \cdot \langle s, a \rangle) \text{ and } g \cdot T(s, a) = T(g \cdot \langle s, a \rangle) \qquad \forall g \in \mathcal{G}_{\mathcal{M}}, s, s' \in \mathbb{S}, a \in \mathbb{A} \tag{5}$$

We refer to a reward function $R$ that satisfies Eq. (4) as a $\mathcal{G}_{\mathcal{M}}$-invariant reward function and a deterministic transition function $T$ that satisfies Eq. (5) as a $\mathcal{G}_{\mathcal{M}}$-equivariant transition function. Note that this is a distinct notion from invariance and equivariance under agent's action in the context of MDP homomorphism. Here, the action refers to the action of a symmetry group, while in MDP homomorphism, the equivariance is to the action of the agent. We use group action or $\mathcal{G}$-action to make this distinction clear when necessary.

For a symmetric MDP that satisfies both Eq. (4) and Eq. (5), both the optimal action-value and optimal policy functions become invariant under $\mathcal{G}_{\mathcal{M}}$ action (Ravindran & Barto, 2001) – that is,

$$Q(s, a) = Q(g \cdot \langle s, a \rangle) \quad \text{and} \quad \pi(a, s) = \pi(g \cdot \langle a, s \rangle) \quad \forall g \in \mathcal{G}_{\mathcal{M}} \ s, a \in \mathbb{S} \times \mathbb{A}. \tag{6}$$

---

[1]This relation is bijective for "simply connected" Lie groups.
[2]We ignore the discount factor for brevity.

The connection of symmetric MDPs to MDP homomorphism is due to the fact that symmetries can be used to define a homomorphism $\mathcal{H} : \mathcal{M} \mapsto \bar{\mathcal{M}}$ by collapsing the state-actions that form an orbit under $\mathcal{G}_\mathcal{M}$. Formally, the collapsed MDP $\bar{\mathcal{M}} = \langle \bar{\mathbb{S}}, \bar{\mathbb{A}}, \bar{R}, \bar{T} \rangle$ is defined by $\bar{\mathbb{S}} = \mathbb{S}/\mathcal{G}_\mathcal{M}$, $\bar{\mathbb{A}} = \mathbb{A}/\mathcal{G}_\mathcal{M}$, $\bar{R}(\langle s, a \rangle^{\mathcal{G}_\mathcal{M}}) = R(s, a)$ and $\bar{T}(s'^{\mathcal{G}_\mathcal{M}} \mid \langle s, a \rangle^{\mathcal{G}_\mathcal{M}}) = T(s' \mid s, a)$. This results in symmetry-based model minimization of symmetric MDPs.

## 4 Desiderata for Symmetry-Based Representation in RL

**Separating Transition and Reward Symmetries**  One important choice is between using the symmetry group of the MDP ($\mathcal{G}_\mathcal{M}$) versus the symmetry group of state-transitions ($\mathcal{G}_T$), where $\mathcal{G}_T$ is the group of transformations of state-action pairs that leads to equivariant deterministic transitions, as given by Equation 5. The former is a subgroup of the latter $\mathcal{G}_\mathcal{M} \leq \mathcal{G}_T$, i.e, the symmetries of a transition model contains the symmetries of the MDP. In fact it is easy to see that $\mathcal{G}_\mathcal{M} = \mathcal{G}_T \cap \mathcal{G}_R$, where $\mathcal{G}_R$ is the group of transformations of state-action pairs that preserve the one step-reward and only satisfy Equation 4. We observe that working with a larger symmetry group $\mathcal{G}_T$ has two benefits: 1) it creates a stronger inductive bias for the model, because in many real-world settings can involve a range of symmetries in transitions that are not present in the reward. For example, an agent's navigation of a 2D map often has the symmetry of the Euclidean group, while the reward (*e.g.*, arriving at a particular location) breaks this symmetry; 2) Separate modeling of transition symmetries facilitates transfer to new tasks, where the reward is changing.

**Invariance/Equivariance in model-free/model-based RL**  If the objective is to carry out model-free RL, Eq. (6) motivates the need to learn action-value functions, or the policies that are *invariant* to symmetries of the MDP ($\mathcal{G}_\mathcal{M}$). For a deterministic policy, the invariance of Eq. (6) becomes an equivariance constraint: $g \cdot \pi(s) = \pi(g \cdot s)$. As it essentially leads to model minimization, van der Pol et al. (2020b); Mondal et al. (2020) use this idea to improve sample efficiency when the groups actions in the agent's action space are known permutations. However, if our objective is just to learn a symmetry-based model of the environment (*i.e.*, transition and reward functions), Eq. (5) suggests that we need to learn a $\mathcal{G}_T$-equivariant transition function.

**Symmetries in a Latent Transition Model**  While it is possible to learn the state transition model in the observation space that is equivariant to the agent's action, for high-dimensional inputs this could be quite challenging since the model has to learn details of the environment that are irrelevant to the RL agent. Using state and action embeddings enables learning of the transition model in the latent space. Indeed the constraint on the model and the embedding is that of the MDP homomorphism Section 3.2. Working in the latent space has an additional benefit when it comes to symmetries: *we can assume that the $\mathcal{G}$ action on the latent state-action pairs is linear through $\rho(\mathcal{G})$ despite having non-linear transformations in the observation space.*

Figure 2: This figure demonstrates the relationship between two types of equivariance in latent variable modeling for an MDP with symmetric transition function. Green arrows (vertical plane) identify a diagram for transition models in an MDP homomorphism. A model $\bar{T}$ and state embedding function $h_\mathbb{S}$ that are equivariant under agent's action makes this diagram commute. Red arrows (horizontal plane) identify the commutativity diagram for a symmetric transition function of an MDP in the latent space. Here the state-action embedding $\langle \tilde{s}, \tilde{a} \rangle$ is produced through the symmetry transformation of another state-action embedding $\langle \bar{s}, \bar{a} \rangle$.

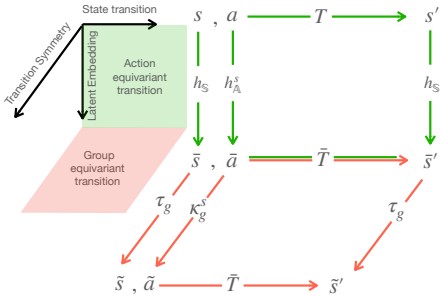

From the fact that symmetries of states $\mathcal{G}_\mathbb{S} \leq \mathcal{G}$ is a subgroup of the state-action or transition symmetry, it follows that $\rho_g \in \rho(\mathcal{G})$ can be divided into two parts: 1) $\tau_g \in \tau(\mathcal{G}_\mathbb{S})$ the group representation acting on the state embedding, and; 2) $\kappa_g^s \in \kappa(\mathcal{G})$, the group representation for state-dependent action embedding.[3]

---

[3]This is because $\rho(\mathcal{G})$ can be seen as a representation that is *induced* by the representation $\tau(\mathcal{G}_\mathbb{S})$ of its subgroup $\rho = \mathrm{Ind}_{\mathcal{G}_\mathbb{S}}^{\mathcal{G}} \tau$.

At this point we can combine the requirement for an MDP homomorphism Eq. (2), with that of the $\mathcal{G}$-equivariant transition model Eq. (5) of a symmetric MDP. The result is the following two constraints in our symmetric latent variable model $\forall s, a \in \mathbb{S} \times \mathbb{A}$ and $g \in \mathcal{G}$ (see Figure 2):

$$\bar{T}(h_{\mathbb{S}}(s), h_{\mathbb{A}}(s,a)) = h_{\mathbb{S}}(T(s,a)) \tag{7}$$

$$\tau_g \bar{T}\left(h_{\mathbb{S}}(s), h_{\mathbb{A}}(s,a)\right) = \bar{T}\left(\tau_g h_{\mathbb{S}}(s), \kappa_g^s h_{\mathbb{A}}(s,a)\right) \tag{8}$$

**Matrix Embedding of States and Actions**   While the design choice of this subsection is not necessary, we see that it can significantly simplify the constraints on a symmetry based model. We propose to use group representations for our state, and state action embeddings $h_{\mathbb{S}} : \mathbb{S} \to \tau(\mathcal{G}_{\mathbb{S}})$ and $h_{\mathbb{A}} : \mathbb{S} \times \mathbb{A} \to \kappa(\mathcal{G})$. This choice assumes that a $\mathcal{G}$ action on state and state-action pairs is *transitive*, so that each state, and state-action pair can be mapped to a (or at least one) group member. To emphasize this in our notation, we use $\kappa(s)$ instead of $h_{\mathbb{S}}(s)$ and similarly use $\tau(s,a)$ instead of $h_{\mathbb{A}}(s,a)$ for state, and state-dependent action embedding respectively. This choice of embedding has several benefits: First, the learned embeddings are automatically equivariant to symmetry transformations of the state, and state-actions:

$$\tau(g \cdot s) = \tau_g \tau(s) \quad \text{and} \quad \kappa(g \cdot \langle s,a \rangle) = \kappa_g^s \kappa(s,a) \quad \forall s, a \in \mathbb{S} \times \mathbb{A}, g \in \mathcal{G}. \tag{9}$$

This means that the symmetries of the state-action pairs are preserved and now take a linear form in the latent space. Note that while the embeddings are automatically equivariant, they may be equivariant to irrelevant non-linear transformations of the input. Word modeling constraints ensure the relevance of these non-linear transformations that are captured by the group equivariant embeddings above. Moreover, this embedding enables the following simple choice for the transition model

$$\bar{T}(\tau(s), \kappa(s,a)) = \kappa(s,a)\tau(s) \tag{10}$$

which simply transforms the state-embedding $\tau(s)$ through the linear group action of state-dependent action encoding $\kappa(s,a)$. Using this transition model, the action equivariance constraint of Eq. (7), and $\mathcal{G}$-equivariance constraint of Eq. (8) simplifies to: $\forall s, a \in \mathbb{S} \times \mathbb{A}$ given a state transition triplet $\{s, a, s'\}$

$$\tau(s') = \kappa(s,a)\tau(s) \tag{11}$$

$$\tau_g \kappa(s,a)\tau(s) = \kappa_g^s \kappa(s,a)\tau_g \tau(s) \tag{12}$$

In practice our model seeks to satisfy these two constraints via the direct minimization of appropriate loss functions, as will be discussed in Section 5.

**Decomposition of the Latent Space**   The decomposition $\mathcal{G} = \mathcal{G}_1 \times \ldots \times \mathcal{G}_K$ into a direct product of subgroups can disentangle the factors of variation in the dataset (Higgins et al., 2018). [4] This gives us a way to represent the latent embedding space as a direct product of $K$ subgroups of $\mathcal{G}$, where each factor varies independently by actions of a subgroup of $\mathcal{G}$ on the latent embedding. Intuitively such a symmetry-based disentanglement provides an effective inductive bias particularly when there is modularity so that temporally coherent changes in the environment are due to the change of a (sparse) subset of factors. We impose this decomposed structure in the form of a direct sum for state representation and the state-dependent action representation – that is $\tau(s) = \bigoplus_k \tau_k(s)$ and $\kappa(s,a) = \bigoplus_k \kappa_k(s,a)$, where $k \in \{1, \ldots, K\}$ and $g = (g_1, \ldots, g_K)$. Moreover the representation of the symmetry group $\mathcal{G}$ acting on the state embedding and the state-dependent action embedding is decomposed as $\tau_g = \bigoplus_k \tau_{g_k}$ and $\kappa_g^s = \bigoplus_k \kappa_{g_k}^s$, where $g_k \in \mathcal{G}_k$ for $\in \{1, \ldots, K\}$ and $g \in \mathcal{G}$. Combining this block structure with the Lie parameterization of Section 3.1 we get

$$\tau_\theta(s) = \bigoplus_k \exp\left(\sum_i \beta_{i,k,\theta}(s)\mathbf{E}^{(i)}\right) \qquad \kappa_\phi(\tau_\theta(s), a) = \bigoplus_k \exp\left(\sum_i \alpha_{i,k,\phi}(\tau_\theta(s), a)\mathbf{E}^{(i)}\right) \tag{13}$$

where we use any standard neural network to implement the $\alpha_\phi$ and $\beta_\theta$ functions above [5]. As we can backpropagate through this function, the network parameters $\theta, \phi$ can be learned end to end. The

---

[4] As noted by Caselles-Dupré et al. (2019), simply having a product structure in the latent space does not guarantee disentanglement, and further constraints are required. In this work, we do not impose any additional constraints for disentanglement.

[5] We denote both, the neural networks which maps to group representations, and network parameters by lowercase greek alphabets.

choice of the subgroup depends on the symmetries of the RL environment, and it only affects the set of bases $\{\mathbf{E}^{(i)}\}_i$ in Eq. (13). For example, in Atari games, the screen often has multiple objects undergoing 2-D translations and rotations, and one can use blocks of the 2-D Special Euclidean ($SE(2)$) that comprise translation and rotations of Euclidean space ($\mathbb{E}$). For more realistic 3D environments, such as those of interest in robotics, self-driving cars and third person games, one can use $SE(3)$, which is the group of 3-D translations and rotations. Also, in theory, we only need specify a group that "contains" the group of interest as a subgroup. For example, if our state-actions only have $90°$ rotational symmetry, we may use a more general group for the representation (*e.g.*, $SE(2)$). The embedding function can define a homomorphism into the relevant subgroup.

## 5 LOSS FUNCTIONS

We consider a standard RL setup where the agent interacts with its environments in episodes and we have access to $(\{s_t, a_t, r_t, s_{t+1}\})_{t=1,..,T}$ where $s_t$ is the state, $a_t$ is the action taken by the agent, $r_t$ is the reward received and $s_{t+1}$ is the observed next state at timestep $t$. Below we describe three loss functions that encode the equivariance/invariance constraints of Eqs. (1), (11) and (12).

**Action Equivariant Transition Loss - Eq. (11)**   Given triplets $\langle s_t, a_t, s_{t+1} \rangle$ from our dataset we simply apply a loss function $\ell$ such as square loss[6] that penalizes the difference between two sides:

$$L_{AET}(\theta, \phi) = \ell\left(\tau_\theta(s_{t+1}), \kappa_\phi(s_t, a_t)\tau_\theta(s_t)\right). \tag{14}$$

The choice of the embedding space and the latent transition function ensure that state embeddings are transformed by linear group action of the action embeddings. Minimization of $L_{AET}$ encourages these symmetry transformations to capture state transitions resulting from the agent's action.

**Group Equivariant Transition Loss - Eq. (12)**   For this we need a $s_{t'}$ in addition to $\langle s_t, a_t \rangle$, where $t'$ can be any state (at different time step in the same or a different episode.) We find the group transformation that maps $s$ to $s'$ the latent space using $\tau_g = \tau_\theta(s_{t'})\tau_\theta(s_t)^{-1}$. Using this we can rewrite Equation Eq. (12) as

$$\underbrace{\tau_\theta(s_{t'})\tau_\theta(s_t)^{-1}}_{\tau_g}\kappa_\phi(s_t, a_t)\tau_\theta(s_t) = \kappa_g^s\kappa_\phi(s_t, a_t)\underbrace{\tau_\theta(s_{t'})}_{\tau_g\tau_\theta(s_t)}. \tag{15}$$

Since the state-dependent action encoding $\kappa_\phi(s_t, a_t)$ for the pair $\langle s_t, a_t \rangle$ is also produced by a neural network, the only missing part in the equation above is $\kappa_g^s$, the state-dependent action transformation. We use a neural network $\rho_\omega : \tau_g \mapsto \kappa_g^s$ to infer it from state transformation $\tau_g$.

**Example 1.** *To get an intuition for what this network is doing consider the example of a pendulum without gravity, with rotation and reflection symmetry $O(2)$ as shown in Fig. 1, where inputs to the networks ($s$) are image sequences and the (ideal) embeddings $\tau_\theta(s), \kappa_\phi(s, a)$ are x-y coordinates plus angular velocity and torque respectively. If we rotate the pendulum using a rotation matrix $\tau_g$, we expect state-dependent action embedding to remain the same since the effect of torque remains similar after rotation. However, if we transform the pendulum by reflection around the vertical axis, we expect that the effect of torque will be negated. $\rho_\omega$ parameterizes this dependence.*

A loss function $\ell$ could then measure the difference between left and the right hand side in the equation above

$$L_{GET}(\theta, \phi, \omega) = \ell\left(\tau_\theta(s_{t'})\tau_\theta(s_t)^{-1}\kappa_\phi(s_t, a_t)\tau_\theta(s_t), \rho_\omega(\tau_\theta(s_{t'})\tau_\theta(s_t)^{-1})\kappa_\phi(s_t, a)\tau_\theta(s_{t'})\right). \tag{16}$$

**Action Invariant Reward Loss - Eq. (1)**   While $L_{AET}$ and $L_{GET}$ enforce the equivariance of the latent transition model to an agent's action and the symmetry group, they do not encode information of the reward in the state representations. In order for the latent model to be homomorphic to the underlying MDP of the environment we match the reward at every state embedding using a reward predictor network $r_\psi : \tau_\theta(s) \mapsto \mathbb{R}$. We measure the difference between the predicted reward and the actual reward at time step $t + 1$:

$$L_R(\psi, \theta, \phi) = \left(r_\psi(\kappa_\phi(\tau_\theta(s_t), a_t)\tau_\theta(s_t)) - r_{t+1}\right)^2. \tag{17}$$

---

[6]In practice we use the normalized square loss $\ell(\boldsymbol{Y}, \boldsymbol{Y}') = \left\|\frac{\boldsymbol{Y}}{\|\boldsymbol{Y}\|_2} - \frac{\boldsymbol{Y}'}{\|\boldsymbol{Y}'\|_2}\right\|_2^2$

## 6 APPLICATION TO MODEL-FREE RL

Following previous success of using transition models for representation learning in model-free RL (Gelada et al., 2019; Schwarzer et al., 2021) we add the losses discussed above to the Temporal Difference (TD) error in Deep Q-learning. In practice, we need to make three modifications to our model/loss.

**Target Network** A trivial solution to both equivariance enforcing losses of Section 5 is to encode all states and actions using an identity matrix. This problem in different contexts is known as the problem of collapse in representation learning. While using the reward signal helps in avoiding the collapse it is often not sufficient specially in sparse reward settings. Following Schwarzer et al. (2021), we use a target network to encode state $s_{t+1}$ and $s_{t'}$ in Eq. (16) in which the network parameters do not receive gradient and moreover are copied from the online network. We explicitly drop the subscripts to differentiate the target from the online network in this section (*e.g.*, $\tau_\theta \to \tau$).

**Projection Head for Transition Losses** Strict enforcement of symmetry constraints by our model can be overly restrictive when the environment has non-symmetric components, or when the our transition model is too simplistic. For this reason, following the previous work, we enforce the losses on a learnable *projection* of the state embedding. That is before application of the loss $\ell$ in Eqs. (14) and (16) we pass the embedding through a projection head.

**M-step prediction** Following the success of (Schwarzer et al., 2021) because of long-term state embedding predictions, we predict state embeddings and rewards for $M$-steps.

### 6.1 PUTTING IT ALL TOGETHER

Considering $M$ consecutive state-actions $\{s_{t:t+M}, a_{t:t+M}\}$ and $\hat{x}_t = x_t = \tau_\theta(s_t)$, we predict the state embeddings and the rewards of next $M$ steps:

$$\hat{x}_{t+m} = \kappa_\phi(\hat{x}_{t+m-1}, a_{t+m-1})\hat{x}_{t+m-1} \quad \text{and} \quad \hat{r}_{t+k} = r_\psi(\hat{x}_{t+k}) \quad \forall m \in \{1, \ldots, M\}$$

Note that we are using $\hat{x}$ for M-step model prediction of the embedding to distinguish them from the latent embedding $x$, and the embedding produced by the target network $\bar{x} = \tau(s_{t+m})$. The same applies to the M-step predicted reward $\hat{r}$ and observed reward $r$. We then project these embedding using a projection head $p_\zeta$ to produce $\hat{z}_{t+m} = p_\zeta(\hat{x}_{t+m})$ and $\bar{z}_{t+m} = p(\bar{x}_{t+m})$. Using this notation, our final expressions for $L_{AET}$ and $L_R$ are:

$$L_{AET} = \sum_{m=1}^{M} \left\| \frac{\hat{z}_{t+m}}{\|\hat{z}_{t+m}\|_2} - \frac{\bar{z}_{t+m}}{\|\bar{z}_{t+m}\|_2} \right\|_2^2 \quad \text{and} \quad L_R = \sum_{m=1}^{M} (\hat{r}_{t+m} - r_{t+m})^2 \tag{18}$$

For $L_{GET}$ we need $\langle s_t, a_t, s_{t+1} \rangle$ and another state $s'$. From their embedding using the notation above we get $\tau_g = \bar{x}_{t'} x_t^{-1}$, the linear transformation between them, and $\kappa_g^s = \rho_\omega(\tau_g)$, the state-dependent action transformation. Now for $\bar{x}_{t'}$, we obtain the predicted next state from $\bar{x}_{t'}$ as $\bar{x}_{t'+1} = \kappa_g^s \kappa(x_t, a_t)\bar{x}_{t'} = \rho_\omega(x_{t'} x_t^{-1})\kappa(x_t, a_t)\bar{x}_{t'}$ and from $\hat{x}_{t+1}$ as $\hat{x}_{t'+1} = \tau_g \hat{x}_{t+1}$. Before penalizing the difference between these embeddings, we project them to $\hat{y}_{t'+1} = b_\eta(\hat{x}_{t'+1})$ and $\bar{y}_{t'+1} = b(\bar{x}_{t'+1})$ using projection head $b_\eta$, to get the final expression for $L_{GET}$:

$$L_{GET} = \left\| \frac{\hat{y}_{t'+1}}{\|\hat{y}_{t'+1}\|_2} - \frac{\bar{y}_{t'+1}}{\|\bar{y}_{t'+1}\|_2} \right\|_2^2 \tag{19}$$

**Q-learning** We pass the representation $x_t$ to a $Q$-learning head $q_\xi$ to learn policies based on the output of the $Q$-value estimator. The $Q$-value estimator is learnt by minimizing:

$$L_{DQN}(\xi, \theta) = (q_\xi(\tau_\theta(s_t), a_t) - (r_t + \gamma \max_a q_\xi(\tau(s_{t+1}), a)))^2 \tag{20}$$

We use the data efficient adaptation of Rainbow (van Hasselt et al., 2019; Hessel et al., 2018) which combines many improvements over the original DQN(Mnih et al., 2013) such as Distributional RL(Dabney et al., 2018), Dueling DQN (Wang et al., 2016), Double DQN (Van Hasselt et al., 2016). The total loss optimized by our model is:

$$L = L_{DQN} + \lambda_1 L_R + \lambda_2 L_{GET} + \lambda_3 L_{AET} \tag{21}$$

where $\lambda_1$, $\lambda_2$ and $\lambda_3$ are hyper-parameters. Motivated by the performance improvements due to augmentation reported in recent literature (Yarats et al., 2021; Schwarzer et al., 2021), we also augment our states by shifting and changing the pixel intensity before encoding them. Fig. 5 in Appendix B.2 shows a detailed schematic of our model. We provide the algorithm for our model in Appendix B.4 and details of network architectures in Appendix B.3.

## 7 EXPERIMENTS

We test our method on a suite of 2D Atari games, which is a popular benchmark used in RL. The full Atari suite consists of 57 games with typically 50 million environment steps. We use the sample-efficient Atari suite introduced by Kaiser et al. (2019), which consists of 26 games with only 100,000 environment steps of training data available. In our experiments, we use three types of simple connected Lie subgroup blocks including General Linear $GL(2)$, Special Euclidean $SE(2)$, and Translation $T(2)$, see Appendix A for more details. Unless stated otherwise, our EqR model uses $SE(2)$ subgroup blocks, with $K = 12$ blocks and $M = 5$ steps during training. $L_{AET}$ is always used to train EqR inorder to model the non-linear transfomations in the state space resulting from agent's actions as linear group actions in the latent space. $L_{GET}$, which makes transition model equivariant with respect to the symmetry transformation of state-actions, and $L_R$ are optional. We build our implementation on top of SPR's (Schwarzer et al., 2021), which is based on `rlpyt` (Stooke & Abbeel, 2019) and PyTorch (Paszke et al., 2019). We use the same underlying RL algorithm and hyperparameters used by SPR for fair comparision.

**Evaluation Metrics**  We compute the average episodic return (the 'game score') at the end of training and normalize it with respect to human scores, as is standard practice. The human-normalized score (HNS) is given by $\frac{\text{agent score - random score}}{\text{human score - random score}}$. Since there is considerable variance across different runs, the mean and the median are not very reliable metrics. Instead, Agarwal et al. (2021) propose using bootstrapped confidence intervals (CI) with stratified sampling which is more suitable for small sample sizes (10 runs per game in our case). We report the Interquartile Mean (IQM), which is the mean across the middle 50% of the runs, as well as the Optimality Gap, which is the amount by which the algorithm fails to meet a minimum HNS of 1.0. We also provide performance profiles showing the fraction of runs above a certain normalized score, which gives a more complete picture of the performance.

**Results**  We use 10 seeds for every game, for every variation of our model. Figure 3 shows performance profiles for two variations of our model, EqR with $L_R$ and with $L_R + L_{GET}$, along with other comparable methods. If one curve is strictly above another, the better method is said to "stochastically dominate" the other (Agarwal et al., 2021). The curves for both variations of the proposed method are almost always above the next best method, SPR (Schwarzer et al., 2021). Figure 4(a) provides results for different methods on all 26 games. The two best variations of the proposed method outperform previous methods, and the difference is statistically significant considering the CI. Table 2 in Appendix B.1 shows full results on all games, and our best model achieves super-human performance on eight games and achieves higher score than any other previous method on 13 out of the 26 games.

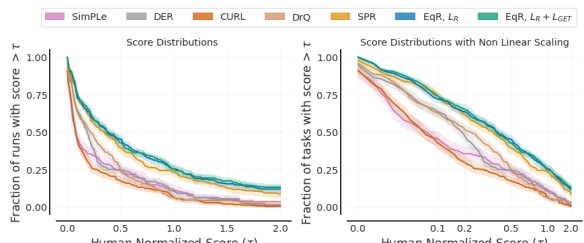

Figure 3: Performance profiles for different methods based on score distributions (left), and average score distributions (right). Shaded regions show pointwise 95% confidence bands. The higher the curve, the better the method is.

In order to better understand the effect of various modeling choices, loss functions and implementation details on the performance, we now consider different variations of EqR, with the same augmentation as the baseline for ablation studies.

**Choice of Group**  To understand the role of the choice of a group in the embedding space, we use our EqR model with $L_R$. This variation of EqR is similar to DeepMDP (Gelada et al., 2019), except for the group structured latent embedding space and group action-based state transition. In order to investigate the effect of the above two group-related constructs, we remove them and use an action

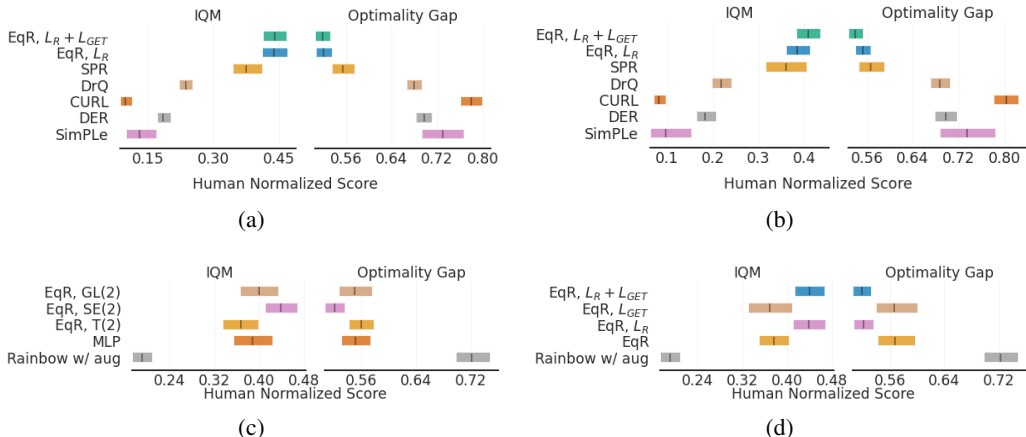

Figure 4: Plots of Interquartile Mean (IQM) and Optimality Gap computed from human-normalized scores, showing the point estimates along with 95% confidence intervals (over 10 runs for all methods, 5 runs for SimPLe). A higher IQM and a lower optimality gap reflects better performance. (a) shows different methods for all 26 games. (b) shows different methods for 17 games. (c) shows the proposed model with different group choices for all 26 games. (d) shows the proposed model with different loss terms for all 26 games.

encoder to predict the next states directly, referring to this as MLP. We further test models with other subgroups including $T(2)$ and $GL(2)$. Figure 4(c) shows that adding a symmetry-based inductive bias in the model by making the embeddings group representations and modeling the transitions as group actions is indeed helpful. The success of the model which uses $SE(2)$ blocks might be attributed to the fact that translations and rotations are the most common types of symmetry transformations present in Atari games. However, the more restrictive $T(2)$ slightly hurts the performance, while the more general $GL(2)$ performs similarly to the MLP model.

**Loss functions** Figure 4(d) compares the performance of EqR using $SE(2)$ subgroup blocks with different loss components. Using EqR with the default $L_{AET}$ results in a considerable improvement over Rainbow with augmentation. Adding $L_{GET}$ improves the performance slightly, while adding only $L_R$ improves the performance even further. We hypothesize that the reward loss is playing a role in both preventing representation collapse and preserving more information of the reward distribution in the latent state embeddings. Adding both $L_{GET}$ and $L_R$ improves the performance only slightly. It might be that this prior of a equivariant transition model with respect to symmetry transformations of state-actions is too restrictive for some games while being beneficial for others. Based on the results in Figure 4(b), in 17 out of a total of 26 games, including this loss term leads to a statistically significant boost in performance. These 17 games are: 'Alien', 'BankHeist', 'BattleZone', 'Boxing', 'ChopperCommand', 'CrazyClimber', 'DemonAttack', 'Freeway', 'Hero', 'Jamesbond', 'MsPacman', 'Pong', 'PrivateEye', 'Qbert', 'RoadRunner', 'Seaquest', 'UpNDown'. The full list of game-wise scores for the ablation studies are presented in Tables 3 and 4 in Appendix B.1.

# 8 CONCLUSION

This paper considers three major symmetry-related constructs within a coherent framework. First, there is the group equivariant state and state-dependent action embedding, which we achieve through Lie parameterization. The world modeling constraints, which are discussed next, further ensure that the transformations captured by the equivariant embedding are relevant. Second, the action equivariant transition, which when combined with group equivariant embeddings, ensures that the state transitions are captured by symmetry transformations in the latent space. Third, there is the group equivariant transition, which acts as an additional bias and ensures that the latent transition model itself is equivariant under symmetry transformations of state-action pairs.

We provided an extensive set of experiments to evaluate the usefulness of our approach in learning state-embeddings for model-free RL. In future work we would like to explore the application of our approach in model-based RL, as well as its ability to generalize across tasks. We also plan to further investigate theoretically grounded methods for combining both symmetric and asymmetric aspects of the environment in the model.

# 9 REPRODUCIBILITY STATEMENT

We have made every effort to ensure that our method is reproducible. Section 5 and Section 6 provide detailed descriptions of the loss functions and various implementation-related details. We present a step-by-step algorithm in Appendix B.4 and have provided details of the network architecture and the list of hyperparameters used, in Sections B.3 and B.5 respectively. Section B.1 has individual results on each game for the proposed EqR model and its variations, to enable researchers to verify the results. Finally, we have submitted the code as part of the submission and we plan on releasing it to the public once the reviewing process is over.

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

## A  SUBGROUP BLOCKS AND THEIR PARAMETERIZATION

### A.1  CHOICE OF GROUP

Atari games differ in their style of play, their objectives, the symmetry transformations of both the agent and other objects on the screen and associated symmetry transformation of the agent's action. But most of these games include symmetry transformations. For example, the screen often has multiple objects undergoing two dimensional(2-D) translations and rotations. In this case one can use blocks of the 2-D Special Euclidean Group $SE(2)$. Each such block can capture the transformation of a particular object in the screen, including the agent. One can also use more restrictive subgroup blocks like $T(2)$, which capture only 2D translations. For more realistic 3D environments, such as those of interest in robotics, self-driving cars and third person games, one can use $SE(3)$, which is the group of 3-D translations and rotations. This should capture both the transformations of the objects in the environment and changes in viewpoint due to the agent's actions.

### A.2  IMPLEMENTING PARAMETERIZATION

We consider three types of subgroups: GL(n) - the set of all invertible linear transformations, SE(n) - the set of all rotations and translations and T(n) - the set of all translations in an n dimensional vector space. We provide a general method to parameterize each of these, based on the type of group.

**GL(n)**   As the matrix representation of GL(n) is the set of invertible matrices which has a measure of 1 it is easy to parameterize it. We just generate $n^2$ parameters using a network corresponding to each element of the matrix. This gives an element from GL(n).

**T(n)**   As T(n) just denotes translation in a n-dimensional space with group action being addition, implementing it is straightforward. We generate $n$ parameters using a neural network and instead of using matrix multiplication use addition for the group action. Note that we can also use a matrix representation for T(n) but it is unnecessary and inefficient.

**SE(n)**   Unlike GL(n) and T(n), parameterizing SE(n) is a bit tricky because it involves parameterizing SO(n). We use a homogeneous co-ordinate based representation of SE(n) = $\left\{ \begin{pmatrix} R & t \\ 0 & 1 \end{pmatrix}, R \in SO(n) \text{ and } t \in T(n) \right\}$. So we need $n$ parameters for the $t$ and another $D = \frac{n(n-1)}{2}$ parameters for $SO(n)$ from the neural network. As explained in Section 4, we can use a Lie parameterization to get the elements of $SO(n)$ by $R = \exp(\sum_{d=1}^{D} \beta_d \mathbf{E}^{(d)})$ where $\mathbf{E}^{(d)}$ denote $D$ bases of the space of skew symmetric matrices and the $\beta_i$s are the parameters of the neural network. For example, in the case of $SO(2)$ we can use the basis $\mathbf{E}^{(1)} = \begin{pmatrix} 0 & 1 \\ -1 & 0 \end{pmatrix}$. Similarly, we can extend this to $SO(n)$ by using a basis given by $D$ $n \times n$ matrices $\mathbf{E}^{(ij)} \; \forall \{1 \le i < j \le n\}$ whose only non-zero elements are $\mathbf{E}_{i,j}^{(ij)} = -1$ and $\mathbf{E}_{j,i}^{(ij)} = 1$.

Although Lie parameterization gives us a general recipe to output a representation of simple connected Lie groups like $SO(n)$, in our implementation we use Euler parameterization because it runs faster in Pytorch. We provide the code for both. Following Quessard et al. (2020), we parameterize each rotation matrix in $SO(n)$ using the product of rotations on $D$ orthogonal planes in $\mathbb{R}^n$: $R = \prod_{i=1}^{n} \prod_{1 \le i < j \le n} R^{ij}$. Here $R^{ij} \in \mathbb{R}^{n \times n}$ is the rotation matrix in the $i - j$ plane, and its non-zero elements besides the diagonal are the four values on the $i, j$ rows and columns, which comprise the 2D rotation matrix that is $R_{i,j}^{ij} = \begin{bmatrix} \cos(\theta_{i,j}) & \sin(\theta_{i,j}) \\ -\sin(\theta_{i,j}) & \cos(\theta_{i,j}) \end{bmatrix}$. We have $D$ parameters $\theta_{i,j}$ which we can obtain from a neural network.

The parameters in all the parameterization techniques mentioned here can be back-propagated. We summarize the number of parameters required from a neural network output, representation type and the associated group actions of different subgroups in Table 1.

Table 1: Group Properties

| Subgroup block type | #Parameters | Representation type | Group action |
|---|---|---|---|
| General Linear - $GL(n)$ | $n^2$ | Matrix$_{n \times n}$ | Matrix multiplication |
| Special Euclidean - $SE(n)$ | $\frac{n(n-1)}{2} + n$ | Matrix$_{(n+1) \times (n+1)}$ | Matrix multiplication |
| Special Orthogonal - $SO(n)$ | $\frac{n(n-1)}{2}$ | Matrix$_{n \times n}$ | Matrix multiplication |
| Translation - $T(n)$ | $n$ | Vector$_n$ | Addition |

# B  ATARI DETAILS

## B.1  FULL RESULTS

We provide individual results on the 26 Atari games after 100K training steps. Our results are averaged over 10 seeds, and the network architectures and full list of hyperparameters used to produce them are provided in Appendix B.3 and Appendix B.5.

- Table 2 compares our two best performing EqR models using $SE(2)$ subgroup blocks with other methods.
- Table 3 compares different choices of subgroup blocks with the reward loss, $L_R$, included for all EqR models (also see Figure 4 (a)).
- Table 4 compares EqR using $SE(2)$ subgroup blocks with different loss terms included in the training objective (see Section 5 and Figure 4 (b)). The action equivariance transition loss, $L_{AET}$ is always included for EqR models.

Table 2: Mean game scores on the 26 Atari games after 100K environment steps. The EqR models use $SE(2)$ subgroup blocks along with an action equivariant transition loss, $L_{AET}$, and are averaged over 10 seeds.

| Game | Random | Human | SimPLe | DER | CURL | DrQ | SPR | EqR, $L_R$ | EqR, $L_R + L_{GET}$ |
|---|---|---|---|---|---|---|---|---|---|
| Alien | 227.8 | 7127.7 | 616.9 | 739.9 | 558.2 | 771.2 | 801.5 | 774.0 | **872.9** |
| Amidar | 5.8 | 1719.5 | 88.0 | **188.6** | 142.1 | 102.8 | 176.3 | 140.9 | 138.4 |
| Assault | 222.4 | 742.0 | 527.2 | 431.2 | 600.6 | 452.4 | 571.0 | **753.8** | 734.3 |
| Asterix | 210.0 | 8503.3 | **1128.3** | 470.8 | 734.5 | 603.5 | 977.8 | 923.2 | 902.5 |
| Bank Heist | 14.2 | 753.1 | 34.2 | 51.0 | 131.6 | 168.9 | 380.9 | 395.1 | **397.4** |
| BattleZone | 2360.0 | 37187.5 | 5184.4 | 10124.6 | 14870.0 | 12954.0 | **16651.0** | 13044.0 | 13255.0 |
| Boxing | 0.1 | 12.1 | 9.1 | 0.2 | 1.2 | 6.0 | 35.8 | 37.5 | **39.2** |
| Breakout | 1.7 | 30.5 | 16.4 | 1.9 | 4.9 | 16.1 | 17.1 | **17.2** | 16.0 |
| ChopperCommand | 811.0 | 7387.8 | **1246.9** | 861.8 | 1058.5 | 780.3 | 974.8 | 1073.5 | 1142.2 |
| Crazy Climber | 10780.5 | 35829.4 | **62583.6** | 16185.3 | 12146.5 | 20516.5 | 42923.6 | 49399.0 | 52008.1 |
| Demon Attack | 152.1 | 1971.0 | 208.1 | 508.0 | 817.6 | **1113.4** | 545.2 | 531.4 | 532.1 |
| Freeway | 0.0 | 29.6 | 20.3 | **27.9** | 26.7 | 9.8 | 24.4 | 24.1 | 25.2 |
| Frostbite | 65.2 | 4334.7 | 254.7 | 866.8 | 1181.3 | 331.1 | 1821.5 | **1855.6** | 1699.4 |
| Gopher | 257.6 | 2412.5 | 771.0 | 349.5 | 669.3 | 636.3 | 715.2 | **1010.0** | 912.1 |
| Hero | 1027.0 | 30826.4 | 2656.6 | 6857.0 | 6279.3 | 3736.3 | **7019.2** | 5775.2 | 6118.5 |
| Jamesbond | 29.0 | 302.8 | 125.3 | 301.6 | **471.0** | 236.0 | 365.4 | 312.8 | 319.7 |
| Kangaroo | 52.0 | 3035.0 | 323.1 | 779.3 | 872.5 | 940.6 | 3276.4 | **3569.3** | 3296.0 |
| Krull | 1598.0 | 2665.5 | 4539.9 | 2851.5 | 4229.6 | 4018.1 | 3688.9 | **5614.5** | 5467.7 |
| Kung Fu Master | 258.5 | 22736.3 | 17257.2 | 14346.1 | 14307.8 | 9111.0 | 13192.7 | **18511.0** | 17510.9 |
| Ms Pacman | 307.3 | 6951.6 | 1480.0 | 1204.1 | 1465.5 | 960.5 | 1313.2 | 1317.1 | **1663.5** |
| Pong | -20.7 | 14.6 | **12.8** | -19.3 | -16.5 | -8.5 | -5.9 | -6.0 | -6.1 |
| Private Eye | 24.9 | 69571.3 | 58.3 | 97.8 | **218.4** | -13.6 | 124.0 | 76.6 | 88.9 |
| Qbert | 163.9 | 13455.0 | **1288.8** | 1152.9 | 1042.4 | 854.4 | 669.1 | 773.8 | 814.9 |
| Road Runner | 11.5 | 7845.0 | 5640.6 | 9600.0 | 5661.0 | 8895.1 | **14220.5** | 13385.0 | 13708.8 |
| Seaquest | 68.4 | 42054.7 | 683.3 | 354.1 | 384.5 | 301.2 | 583.1 | 650.3 | **697.9** |
| Up N Down | 533.4 | 11693.2 | 3350.3 | 2877.4 | 2955.2 | 3180.8 | 28138.5 | 44295.4 | **52118.4** |
| Mean Human-Norm'd | 0.000 | 1.000 | 0.443 | 0.285 | 0.381 | 0.357 | 0.704 | 0.859 | **0.886** |
| Median Human-Norm'd | 0.000 | 1.000 | 0.144 | 0.161 | 0.175 | 0.268 | 0.415 | **0.418** | 0.398 |
| # Superhuman games | 0 | N/A | 2 | 2 | 2 | 2 | 7 | **8** | 7 |

Table 3: Mean game scores on the 26 Atari games after 100K environment steps for different choices of subgroup blocks, averaged over 10 seeds. The reward loss, $L_R$, is included in addition to the default loss $L_{AET}$.

| Game | Random | Human | MLP | EqR, $T_2$ | EqR, $SE_2$ | EqR, $GL_2$ |
|---|---|---|---|---|---|---|
| Alien | 227.8 | 7127.7 | 780.1 | 846.4 | 774.0 | **881.3** |
| Amidar | 5.8 | 1719.5 | **143.3** | 139.7 | 140.9 | 132.2 |
| Assault | 222.4 | 742.0 | 701.5 | 684.0 | **753.8** | 692.3 |
| Asterix | 210.0 | 8503.3 | 973.6 | **1004.4** | 923.2 | 889.5 |
| Bank Heist | 14.2 | 753.1 | 402.1 | 353.5 | 395.1 | **430.2** |
| BattleZone | 2360.0 | 37187.5 | 12722.6 | 11500.0 | 13044.0 | **13114.0** |
| Boxing | 0.1 | 12.1 | **38.0** | 28.9 | 37.5 | 33.4 |
| Breakout | 1.7 | 30.5 | 16.2 | 14.8 | **17.2** | 15.4 |
| ChopperCommand | 811.0 | 7387.8 | 989.5 | 1028.9 | 1073.5 | **1088.1** |
| Crazy Climber | 10780.5 | 35829.4 | 43705.8 | 50822.1 | 49399.0 | **55018.5** |
| Demon Attack | 152.1 | 1971.0 | 518.6 | **544.2** | 531.4 | 510.2 |
| Freeway | 0.0 | 29.6 | 20.3 | 18.5 | **24.1** | 21.4 |
| Frostbite | 65.2 | 4334.7 | 1702.4 | 1653.8 | **1855.6** | 1797.7 |
| Gopher | 257.6 | 2412.5 | 720.2 | **1012.5** | 1010.0 | 894.4 |
| Hero | 1027.0 | 30826.4 | **6840.0** | 5779.8 | 5775.2 | 5934.7 |
| Jamesbond | 29.0 | 302.8 | **337.4** | 313.25 | 312.8 | 334.8 |
| Kangaroo | 52.0 | 3035.0 | 2994.8 | 2942.5 | **3569.3** | 3186.4 |
| Krull | 1598.0 | 2665.5 | 3801.5 | 5293.0 | 5614.5 | **5772.6** |
| Kung Fu Master | 258.5 | 22736.3 | 13780.4 | 14924.2 | **18511.0** | 16002.8 |
| Ms Pacman | 307.3 | 6951.6 | 1220.8 | 1166.8 | **1317.1** | 1147.7 |
| Pong | -20.7 | 14.6 | -6.1 | -11.5 | **-6.0** | -8.2 |
| Private Eye | 24.9 | 69571.3 | 72.4 | 65.1 | **76.6** | 55.9 |
| Qbert | 163.9 | 13455.0 | 678.4 | 763.2 | **773.8** | 635.2 |
| Road Runner | 11.5 | 7845.0 | 12765.2 | **13654.2** | 13385.0 | 12560.4 |
| Seaquest | 68.4 | 42054.7 | **656.9** | 647.3 | 650.3 | 633.3 |
| Up N Down | 533.4 | 11693.2 | 23130.6 | **58164.4** | 44295.4 | 43767.2 |
| Mean Human-Norm'd | 0.000 | 1.000 | 0.681 | 0.829 | **0.859** | 0.833 |
| Median Human-Norm'd | 0.000 | 1.000 | 0.398 | 0.361 | **0.418** | 0.380 |
| # Superhuman games | 0 | N/A | 6 | 6 | **8** | 7 |

Table 4: Mean game scores on the 26 Atari games after 100K environment steps for EqR using $SE(2)$ subgroup blocks with different loss terms included in the training objective. The action equivariance transition loss, $L_{AET}$, is included for all EqR models and the scores are averaged over 10 seeds.

| Game | Random | Human | EqR | EqR, $L_R$ | EqR, $L_{GET}$ | EqR, $L_R + L_{GET}$ |
|---|---|---|---|---|---|---|
| Alien | 227.8 | 7127.7 | 856.5 | 774.0 | 862.5 | **872.9** |
| Amidar | 5.8 | 1719.5 | 134.7 | **140.9** | 135.0 | 138.4 |
| Assault | 222.4 | 742.0 | 643.1 | **753.8** | 701.3 | 734.3 |
| Asterix | 210.0 | 8503.3 | 824.8 | **923.2** | 864.9 | 902.5 |
| Bank Heist | 14.2 | 753.1 | **407.3** | 395.1 | 335.9 | 397.4 |
| BattleZone | 2360.0 | 37187.5 | 12805.6 | 13044.0 | 12990.4 | **13255.0** |
| Boxing | 0.1 | 12.1 | 32.7 | 37.5 | 34.8 | **39.2** |
| Breakout | 1.7 | 30.5 | 14.6 | **17.2** | 14.8 | 16.0 |
| ChopperCommand | 811.0 | 7387.8 | 1015.6 | 1073.5 | 934.8 | **1142.2** |
| Crazy Climber | 10780.5 | 35829.4 | 38483.9 | 49399.0 | 43085.6 | **52008.1** |
| Demon Attack | 152.1 | 1971.0 | 523.8 | 531.4 | 504.6 | **532.1** |
| Freeway | 0.0 | 29.6 | 22.1 | 24.1 | 22.5 | **25.2** |
| Frostbite | 65.2 | 4334.7 | 1635.2 | **1855.6** | 1563.9 | 1699.4 |
| Gopher | 257.6 | 2412.5 | 695.3 | **1010.0** | 789.3 | 912.1 |
| Hero | 1027.0 | 30826.4 | 5763.9 | 5775.2 | 5603.8 | **6118.5** |
| Jamesbond | 29.0 | 302.8 | **388.4** | 312.8 | 344.9 | 319.7 |
| Kangaroo | 52.0 | 3035.0 | 2667.9 | **3569.3** | 2848.7 | 3296.0 |
| Krull | 1598.0 | 2665.5 | 4209.2 | **5614.5** | 4411.2 | 5467.7 |
| Kung Fu Master | 258.5 | 22736.3 | 12287.9 | **18511.0** | 16394.6 | 17510.9 |
| Ms Pacman | 307.3 | 6951.6 | 1141.3 | 1317.1 | 1514.7 | **1663.5** |
| Pong | -20.7 | 14.6 | -9.9 | **-6.0** | -6.5 | -6.1 |
| Private Eye | 24.9 | 69571.3 | 73.2 | 76.6 | 87.5 | **88.9** |
| Qbert | 163.9 | 13455.0 | 696.7 | 773.8 | 736.8 | **814.9** |
| Road Runner | 11.5 | 7845.0 | 12659.2 | 13385.0 | 13110.4 | **13708.8** |
| Seaquest | 68.4 | 42054.7 | 593.6 | 650.3 | 641.0 | **697.9** |
| Up N Down | 533.4 | 11693.2 | 29425.4 | 44295.4 | 39076.6 | **52118.4** |
| Mean Human-Norm'd | 0.000 | 1.000 | 0.682 | 0.859 | 0.749 | **0.886** |
| Median Human-Norm'd | 0.000 | 1.000 | 0.337 | **0.418** | 0.377 | 0.398 |
| # Superhuman games | 0 | N/A | 6 | **8** | 6 | 7 |

B.2    MODEL SCHEMATIC

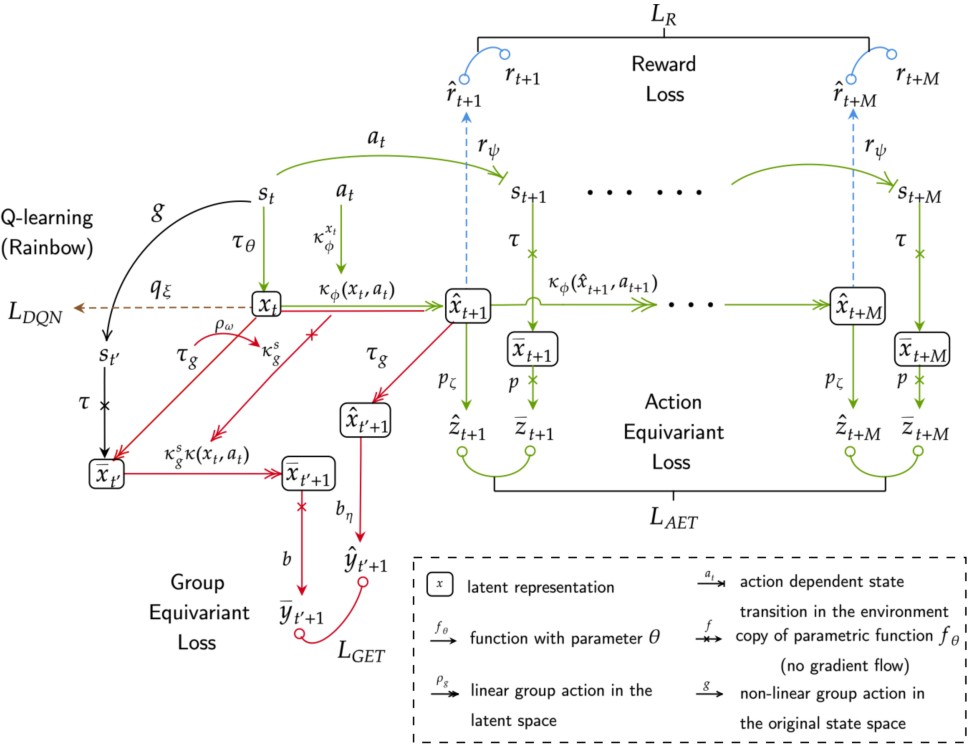

Figure 5: A schematic of the EqR model, applied to model-free RL. Green in the framework corresponds to learning equivariance under the agent's action and red corresponds to learning equivariance of the transition model with respect to symmetry transformation of the state-action. This color scheme is consistent with Figure 2. The part of the framework that corresponds to reward matching and Q-learning is shown in blue and brown respectively. The arrows in the schematic are differentiated by their heads and are described in the legend.

B.3    NETWORK ARCHITECTURE

We follow the baseline RL implementation of DrQ (Yarats et al., 2021) and SPR (Schwarzer et al., 2021) by using the 3-layer convolutional encoder from (Mnih et al., 2015) and then use a linear layer to get the parameters for the Group Parameterization. The output size of this layer varies depending on the group type, the number of blocks used and the size of the group. This defines our $\tau_\theta$. Note that the output of our encoder is a matrix for $GL(n)$ and $SE(n)$. We flatten it before we feed to other neural network like the Q-head $q_\xi(\cdot)$.

For the action encoder $\kappa(\cdot)$ we use a simple 1 layer MLP with batchnorm, ReLU and a hidden size of 256. We concatenate the one-hot encodings of the actions with the state representations coming from $\tau_\theta$ and pass it through the action encoder to get matrix representation of the group after parameterization.

For the reward predictor network $r_\psi$ we use a 2-layered MLP with batchnorm, ReLU and a hidden size of 256.

For the Q-head $q_\xi(\cdot)$ we use 2-layered MLP as well.

For the projection head $p_\zeta(\cdot)$ we share the first layer of Q-head whereas for projection head $b_\eta(\cdot)$ we use a single layer MLP.

## B.4 ALGORITHM

---

**Algorithm 1:** Equivariant Representations for RL

---

Denote the parameters of online networks $\tau_\theta, \kappa_\phi, p_\zeta, b_\eta$ as $\Theta_o$
Denote the parameters of target networks $\tau, \kappa, p, b$ as $\Theta_c$
Denote the parameters of networks $\rho_\omega, q_\xi$ as $\Phi$
Denote the dept of the prediction as $M$ and batch size as $N$
Initialize the replay buffer $\mathcal{B}$
**while** *Training* **do**

    Collect $\{s, a, r, s'\}$ using policy with $(\Theta_o, \Phi)$ and add to the buffer $\mathcal{B}$
    Sample a minibatch of $M$ length sequences $\{s_{0:M}, a_{0:M}, r_{0:M}\} \sim \mathcal{B}$
    **for** $i$ *in range*$(0, N)$ **do**

        **if** *augmentation* **then**
            $s_{0:M}^i \leftarrow augment(s_{0:M}^i)$
        **end**

        $x_0^i \leftarrow \tau_\theta(s_0^i);$                          // state representation
        $\hat{x}_0^i \leftarrow x_0^i$
        $l^i \leftarrow 0$
        **for** $k$ *in range*$(1, M+1)$ **do**

            $\hat{x}_k^i \leftarrow \kappa_\phi(\hat{x}_{k-1}^i, a_{k-1}^i)\hat{x}_{k-1}^i;$    // state transition by group action
            $\bar{x}_k^i \leftarrow \tau(s_k^i);$                 // target state representation
            $\hat{z}_k^i \leftarrow p_\zeta(\hat{x}_k^i), \bar{z}_k^i \leftarrow p_\zeta(\bar{x}_k^i);$              // projections
            $l^i \leftarrow l^i + \lambda_2 \|\frac{\hat{z}_{t+k}}{\|\hat{z}_{t+k}\|_2} - \frac{\bar{z}_{t+k}}{\|\bar{z}_{t+k}\|_2}\|_2^2;$    // compute $L_{AET}$ at step k
            $\hat{r}_k^i \leftarrow r_\psi(\hat{x}_k^i);$                 // predict rewards
            $l^i \leftarrow l^i + \lambda_1 \|\hat{r}_k^i - r_k^i\|_2^2;$         // compute $L_R$ at step k
        **end**

        $j \sim \{0, .., N-1\};$             // uniformly sample an index
        $\bar{x}_0^j \leftarrow \tau(s_0^j);$  // encode the state for that index from the batch
        $\tau_g^i = \bar{x}_0^j {x_0^i}^{-1};$             // find the group representation
        $\hat{x}_1^j \leftarrow \tau_g^i \hat{x}_1^i;$              // next state by group action
        $\bar{x}_1^j \leftarrow \rho_\omega(\tau_g^i)\kappa(x_0^i, a_0^i)\bar{x}_0^j;$     // next state by action-embedding
        $\hat{y}_1^i \leftarrow b_\eta(\hat{x}_1^j), \bar{y}_1^i \leftarrow b(\bar{x}_1^j);$            // projections
        $l^i \leftarrow l^i + \lambda_3 \|\frac{\hat{y}_1^i}{\|\hat{y}_1^i\|_2} - \frac{\bar{y}_1^i}{\|\bar{y}_1^i\|_2}\|_2^2;$           // compute $L_{GET}$
        $l^i \leftarrow l^i + RLloss(\hat{x}_0^i, a_0^i, r_0^i, \bar{x}_1^i; q_\xi)$
    **end**

    $l \leftarrow \frac{1}{N}\sum_{i=0}^N l_i;$                 // average over minibatch
    $\Theta_o, \Phi \leftarrow optmize((\Theta_o, \Phi), l);$         // update online networks
    $\Theta_c \leftarrow \Theta_o;$           // copy weights to target networks
**end**

---

## B.5 HYPERPARAMETERS

In this section, we provide the full set of hyperparameters in our model. As mentioned earlier, our baseline RL algorithm closely follows SPR's (Schwarzer et al., 2021) implementation of Rainbow and hence we use most of their hyperparameters setting in order to be able to compare to them. Note that the weights of $L_R$ - $\lambda_1$, $L_{AET}$ - $\lambda_2$ and $L_{GET}$ - $\lambda_3$ are set to one whenever they are used in the model.

Table 5: Hyperparameters for ErQ (including variations) on Atari.

| Parameter | Setting |
|---|---|
| Gray-scaling | True |
| Observation down-sampling | $84 \times 84$ |
| Frames stacked | 4 |
| Action repetitions | 4 |
| Reward clipping | $[-1, 1]$ |
| Terminal on loss of life | True |
| Max frames per episode | 108K |
| Update | Distributional Q |
| Dueling | True |
| Support of Q-distribution | 51 |
| Discount factor | 0.99 |
| Minibatch size | 32 |
| Optimizer | Adam |
| Optimizer: learning rate | 0.0001 |
| Optimizer: $\beta_1$ | 0.9 |
| Optimizer: $\beta_2$ | 0.999 |
| Optimizer: $\epsilon$ | 0.00015 |
| Max gradient norm | 10 |
| Priority exponent | 0.5 |
| Priority correction | $0.4 \to 1$ |
| Exploration | Noisy nets |
| Noisy nets parameter | 0.5 |
| Training steps | 100K |
| Evaluation trajectories | 100 |
| Min buffer size for sampling | 2000 |
| Replay period every | 1 step |
| Updates per step | 2 |
| Multi-step return length | 10 |
| Prediction depth, $M$ | 5 |
| $\lambda_1$ | 1 |
| $\lambda_2$ | 1 |
| $\lambda_3$ | 1 |
| Data Augmentation | Random shifts (±4 pixels) Intensity(scale=0.05) |

| Parameter | Setting ($T_2$) | Setting ($SE_2$) | Setting ($GL_2$) |
|---|---|---|---|
| Num Blocks (K) | 32 | 12 | 12 |
| Group Action | Addition | MatMul | MatMul |

