# OpenReview forum: "EqR: Equivariant Representations for Data-Efficient Reinforcement Learning"
_ICLR.cc/2022/Conference — ICLR 2022 Submitted_

### Official Review · Reviewer_dVfB · 2021-10-30

**Correctness:** 2
**Technical Novelty And Significance:** 1
**Empirical Novelty And Significance:** 2
**Recommendation:** 5
**Confidence:** 3

**Main Review:**

### Strengths

- The paper is well-motivated and addresses an interesting and important problem.
- The idea of embedding states and actions as matrices representing a group action is intriguing, and may induce interesting properties in the representation beyond those studied in this paper.
- The paper provides sufficient background to be understood by readers with some but not extensive experience in the topic.


### Weaknesses

- **The justification for the proposed method is lacking** — section 4 provides a number of desiderata, but there is no follow-up showing empirically or theoretically that the resulting method (especially given the use of target networks and projections) will actually satisfy these desiderata.
- **The use of Lie groups is not motivated in Section 3**. As a reader, I presumed that this class was chosen in order to justify the use of learned matrix embeddings in the proposed model, but this would be worth clarifying. Additionally, the basis of the Lie algebra is not referred to in Section 6 despite an indication that it will be used in Section 3.

- **The paper suffers from a lack of clarity.** For example, in the discussion of symmetric MDPs the group action is applied to both states and state-action pairs, but it is not clear whether these represent two separate actions, or whether $g\cdot \langle s, a \rangle$ has the same effect on states as $g \cdot s$. Additionally, the notation $\kappa (\mathcal{G})$ and $\tau(\mathcal{G}_{\mathbb{S}})$ is not clear on a first read. In general I think more explicitly stating the types of the mathematical objects used in the paper will make it much easier to parse. By default, I and probably many other RL readers default to assuming vector-valued embeddings, and it is easier to override this default when the output type of the embedding is made explicit.
	- Section 4 is quite long and in many cases it’s not clear whether the proposed properties are _desiderata_ or guaranteed to be satisfied by the proposed method. For example, the matrix embeddings of state actions seems to hold by construction but disentanglement presumably does not.
- **The empirical evaluations don’t support the main goal of the paper.** Adding the group-equivariant loss doesn’t provide much improvement over the standard model-learning loss — particularly when we look at median performance. This suggests that most of the benefit over the baselines is coming from the model-learning component of the objective rather than the incorporation of symmetries. Further, the method without the symmetry-learning loss is similar to many existing representation learning methods such as DeepMDP, PSR, PBL, MuZero, LatCo, and DBC. The equivariant loss is therefore critical to the novelty of the proposed algorithm, yet it doesn’t seem to have a significant impact on overall performance. It is possible that a more careful empirical analysis on environments with more explicit symmetries may reveal that this loss is indeed useful, but the current experimental results are insufficient to show this to a satisfactory degree.
    - The experimental results are limited. The paper only uses one evaluation scheme and doesn’t provide much insight into whether the learned model really is equivariant to the set of transformations used.
    - Further, the baselines used in the paper don’t seem to be described or cited, which makes it difficult to determine whether a fair comparison is really being made between the different methods. The baselines I was already familiar with mostly seem to make use of model-free approaches combined with data augmentation, which leaves the model-learning component of EqR as a confounding factor. SPR, which does include a learned latent-space model, seems to attain a higher median human-normalized score than EqR with the equivariant transition loss, suggesting that at least part of the improved performance is coming from the specific parameterization of the model.

### Concrete avenues for improvement

- I am very intrigued by the use of matrix embeddings in the representation — typically in RL we treat representations as vectors, and it would be interesting to see whether enforcing linear actions of the transition and symmetry operators on the latent space induces particular structure.
- It is possible that by looking at environments with more structure, the proposed method might be shown to capture the appropriate in/equivariances and therefore improve performance and sample efficiency. I recommend incorporating evaluations on e.g. Mujoco or some other environment with more explicit physically-based invariances.
- The focus of this paper is on equivariant representations for sample efficiency; however, it seems like an arguably more promising application is in improving generalization.
- Ensuring that symbols and their types are defined before they are used will make the paper much more readable.
- Section 4 should be rewritten to more clearly express which of the desiderata are definitely satisfied by the proposed method, and which are not guaranteed to be satisfied. It might also be interesting to include empirical investigations into the degree to which the proposed method satisfies these desiderata.

**Summary Of The Paper:**

The paper focuses on the problem of incorporating symmetries into RL agents’ representations to improve data efficiency. It proposes a model-based representation learning method parameterized by a group action that is based on encoding states and state-action pairs as elements of the group's representation in latent space. The proposed method is evaluated against a number of baselines in the data-limited Atari setting.

**Summary Of The Review:**

I recommend rejecting this paper due to a) the limited empirical benefit of incorporating equivariance into the proposed representation learning method and b) the lack of clarity in both the paper’s exposition and its claims about the proposed method’s properties. With additional work and a more informative evaluation setting, I think the paper will be in a much better position for consideration in future conferences.

---

> ### Author Response · Authors · 2021-11-19
> **Response to Reviewer dVfB (Part 1/2)**
>
> We thank the reviewer for their in-depth comments and for their appreciation of the basic idea behind the paper.
>
> > The justification for the proposed method is lacking
>
> We presented the desiderata in order to theoretically motivate all choices in the proposed method. However, the application of our method to model-free RL requires some adjustments, which are discussed in Section 6. The Lie algebraic parameterization used in our framework guarantees that the neural network encodes states as group element representations. The other desiderata, except for disentanglement, are enforced through the individual loss terms. We have discussed in Section 6, “Projection Head for Transition Losses”, that our symmetry constraints can be overly restrictive if the environment has highly non-symmetric components or if our transition model is too simplistic. For this reason, we enforce the losses on a learnable projection of the state embeddings. Regarding disentanglement, we have clarified in Section 4, “Decomposition of the Latent Space”, that the use of a product structure in the latent space encourages disentanglement but does not guarantee it (as shown in [4]).
>
> > The use of Lie groups is not motivated in Section 3
>
> We have clearly motivated the use of Lie groups in Section 3. As stated in this section, we use Lie groups because they can capture both continuous and finite symmetry transformations (as subgroups) and show that there is a straightforward way to parameterize them by choosing the basis of the Lie Algebra. This construction guarantees that the neural network encodes states as group element representations. We thank the reviewer for pointing out an error in Section 3. We now direct the reader to Section 4 where we refer to this parameterization (and not erroneously to Section 6). Since choosing a particular basis is application dependent, we postpone this choice for our particular implementation in Section 4, to keep the theory and the ideas of the paper general.
>
> > The paper suffers from a lack of clarity
>
> Whereas we provide some background on group theory, as it relates to our paper, readers without any background in equivariant deep learning could indeed find the paper a bit challenging to follow. In the revised version, we have tried to improve the exposition throughout.
>
> We answer specific questions in the comments below.
>
> > For example, in the discussion of symmetric MDPs the group action is applied to both states and state-action pairs, but it is not clear whether these represent two separate actions, or whether $g\cdot\langle s,a\rangle$ has the same effect on states as $g\cdot s$.
>
> It is common to denote group action using abstract notation for simplicity. In the particular example above, the same group element $g$ can act on both states in $\mathbb{S}$ and state-actions in $\mathbb{S \times A}$, where the group action on $s$ is induced by the action on the pair $\langle s, a \rangle$. For example, in Figure 1 with the pendulum example, if the group corresponds to reflection about the vertical axis, then the action of $g$ on a state reflects the pendulum, and its action on the state-action pair reflects the pendulum (state) as well as permutes the action.
>
> > Additionally, the notation $\kappa(\mathcal{G})$ and $\tau(\mathcal{G}_\mathbb{S})$ is not clear on a first read.
>
> Given that one of our contributions is the use of Lie group representations for the state and action embeddings, we have clearly defined $\kappa(\mathcal{G})$ and $\tau(\mathcal{G}_\mathbb{S})$ as group representations before using them (see “Matrix Embedding of States and Actions” paragraph in Section 4).
>
> Additionally, we have made several changes to improve the readability of the paper. We have added a new diagram (Figure 2) that corresponds closely with the theory described in Section 3 and Section 4, and have moved the color implementation-specific diagram (Figure 5) to the appendix. We have also added details on the choice of groups (Section 4, “Decomposition of the Latent Space”) to provide intuition on the type of symmetry we try to capture in Atari games. We have also expanded on several explanations throughout the paper, which the reader might find helpful.

---

> ### Author Response · Authors · 2021-11-19
> **Response to Reviewer dVfB (Part 2/2)**
>
> Regarding the empirical evaluations, we respectfully challenge some of the reviewer’s claims.
>
> > This suggests that most of the benefit over the baselines is coming from the model-learning component of the objective rather than the incorporation of symmetries. Further, the method without the symmetry-learning loss is similar to many existing representation learning methods such as DeepMDP, PSR, PBL, MuZero, LatCo, and DBC.
>
> In the original paper, we had specifically designed an experiment to investigate the
> effect of the symmetry-related components, by removing them and using an MLP action encoder as the model-learning component. We have already clearly stated that this MLP setting of our model is similar to DeepMDP [1], and have empirically shown that adding the symmetry components leads to better performance. We now strengthen this claim by providing more reliable evaluation metrics in our revised version (Section 7 “Choice of Group”).
>
> > The equivariant loss is therefore critical to the novelty of the proposed algorithm, yet it doesn’t seem to have a significant impact on overall performance.
>
> First, we would like to point out that the state representations learned by our model can be equivariant even without the group-equivariant loss, $L_{GET}$. The reviewer’s claim above reflects a misunderstanding of the contributions of the paper and the different ways in which symmetry is leveraged in our approach. We direct the reviewer to “A General Note on Losses and Symmetries” in the general comment. Second, we have now performed a more comprehensive analysis using more reliable metrics, as suggested in recent work [2]. These results and the relevant discussion, which are provided in the revised paper (Section 7, “Loss functions”), demonstrate that including $L_{GET}$ in the training objective leads to an improvement in a majority of the games.
>
> > Further, the baselines used in the paper don’t seem to be described or cited, which makes it difficult to determine whether a fair comparison is really being made between the different methods.
>
> Thank you for pointing this out, we were missing the name of one of the papers in the related works section (the citation was still present). All other methods were already named and cited clearly (see Section 2).
>
> > SPR, which does include a learned latent-space model, seems to attain a higher median human-normalized score than EqR with the equivariant transition loss, suggesting that at least part of the improved performance is coming from the specific parameterization of the model.
>
> We believe that we have already addressed this concern of the reviewer regarding the benefit of including the group-equivariant loss, $L_{GET}$.
>
> A few other general concerns about the statistical significance of the empirical results and the visualization of the learned latent space, are addressed in the general comment above.
>
> *References:*
>
> [1] Gelada, C., Kumar, S., Buckman, J., Nachum, O. and Bellemare, M.G., 2019, January. DeepMDP: Learning Continuous Latent Space Models for Representation Learning. In ICML.
>
> [2] Agarwal, R., Schwarzer, M., Castro, P.S., Courville, A. and Bellemare, M.G., 2021. Deep reinforcement learning at the edge of the statistical precipice. In NeurIPS.
>
> [3] Schwarzer, M., Anand, A., Goel, R., Hjelm, R.D., Courville, A. and Bachman, P., 2020, September. Data-Efficient Reinforcement Learning with Self-Predictive Representations. In International Conference on Learning Representations.
>
> [4] Caselles-Dupré, H., Garcia Ortiz, M. and Filliat, D., 2019. Symmetry-based disentangled representation learning requires interaction with environments. In NeurIPS.

---

> ### Author Response · Authors · 2021-11-22
> **Followup on Response to Reviewer dVfB**
>
> As today is the last day of the discussion period, we would like to know if the reviewer needs any further clarifications on the paper or our comments.
>
> We will also like to reiterate that we have added a lot more clarifications on the type of symmetry constructs used in this work throughout the revised paper to improve its readability. We have now shown the benefit of modeling symmetry in the latent space using more statistically reliable evaluation metrics.

---

> > ### Comment · Reviewer_dVfB · 2021-11-28
> > **Clarity has improved, but concerns about technical contribution remain**
> >
> > Thanks to the authors for their detailed response, and I apologize for the delay as it took me some time to be able to review the updated paper in depth and consider the responses to my initial review. The updates to the paper have significantly improved its clarity and I appreciate the updated experiments section. Unfortunately, I still think that the current framing of the contribution is not consistent with the experimental results. In particular:
> >
> > - As noted previously, it doesn’t seem that the $L_{GET}$ loss is significantly contributing to the data efficiency gain over SPR. EQR with only $L_{AET}$ similarly doesn’t produce a noticeable performance gain over SPR, suggesting that it is in fact the reward-invariant loss that is contributing to the improvement in the IQM results shown in Figure 4. However, the combination of $L_{AET}$ and $L_R$ strongly resembles prior work on learned latent space models such as DeepMDP. EqR certainly has some differences, notably a different parameterization of the latent representations, an assumed linearity in the transition model, and a state-specific action embedding, however given the nature of the evaluation environments (discussed in the following bullet point), it’s not clear to me whether these different design choices are significant to the performance of the method.
> >
> > - The evaluation domain is not suited to evaluate the contributions of the proposed method. Atari does not contain natural sets of symmetries that EqR is designed to learn, so it is impossible to verify whether the learned latent space symmetries correspond to anything meaningful in the environment. Further, the transitions seem to only be trivially action-equivariant:  the “up” action always takes the agent up, while the “down” action similarly always takes it down. There aren’t any situations to my knowledge that resemble the example provided on page 6. This is likely why $L_{AET}$ alone doesn’t see a significant improvement over SPR and why adding $L_{GET}$ similarly doesn’t yield a performance improvement.
> >
> > Overall, I think the paper has some promising ideas, but it really needs a deeper analysis both in terms of the effect of the proposed latent space parameterization on representation learning, and evaluations on environments that have the type of structure the method seems designed to learn. As it stands, the aspects of the method that distinguish it from prior work don't seem to significantly influence performance, limiting the significance and novelty of the resulting contribution.

---

> > > ### Author Response · Authors · 2021-11-28
> > > **Response on technical contributions to Reviewer dVfB (Part 1/2)**
> > >
> > > We thank the reviewer for their response. We politely disagree with the reviewer on the above points and believe there are still some misunderstandings.
> > >
> > > > As noted previously, it doesn’t seem that the $L_{GET}$ loss is significantly contributing to the data efficiency gain over SPR.
> > >
> > > We believe we have already addressed this concern in the previous response and in the revised paper (e.g., see Figure 4(b)). We urge the reviewer to revisit our general comment, specifically the sections “Symmetric Transition Loss Term” and “A General Note on Losses and Symmetries”.
> > >
> > > > EQR with only $L_{AET}$ similarly doesn’t produce a noticeable performance gain over SPR, suggesting that it is in fact the reward-invariant loss that is contributing to the improvement in the IQM results shown in Figure 4. However, the combination of $L_{AET}$ and $L_R$ strongly resembles prior work on learned latent space models such as DeepMDP. EqR certainly has some differences, notably a different parameterization of the latent representations, an assumed linearity in the transition model, and a state-specific action embedding, however given the nature of the evaluation environments (discussed in the following bullet point), it’s not clear to me whether these different design choices are significant to the performance of the method.
> > >
> > > We have an experiment in the paper that exactly addresses your point. We have used a model which we call ‘MLP’ (see Section 7 “Choice of Group”, Figure 4(c)), where we remove all our group-based formulations but keep $L_R$ and $L_{AET}$ for fair comparison. This model is similar to DeepMDP [1] with normalized MSE for latent transition ($L_{AET}$ in our setup) and SPR [2] with the reward loss $L_R$. In fact, SPR is similar to DeepMDP except for the reward loss and the choice of loss for latent transition - DeepMDP uses MSE loss for latent transitions while SPR uses negative cosine similarity (as written in the paper, but in the actual implementation SPR uses normalized MSE similar to BYOL [3] to deal with representation collapse). We have clearly shown that our method with the group-based formulation outperforms SPR, see Figure 4 (a), and the above-described ‘MLP’ model (which covers both DeepMDP and SPR with reward loss included, see Figure 4 (c)). This suggests that the improvement is not from the loss functions (i.e., $L_{AET}$ and $L_R$) themselves but the particular structure of the latent space and the transitions, which, to the best of our knowledge, has not been considered in prior works in the RL literature.
> > >
> > > We believe the reward-invariant loss term is playing a crucial role in preventing the representation collapse of some states. Due to $L_{AET}$ (along with the structure of the latent space and the transitions), the model can choose to only focus on the transformations resulting from the agent’s action and ignore the transformations in the rest of the environment. Including the reward loss encourages the latent space to preserve the reward information in different states and hence forces the model to learn other relevant transformations in the environment. Therefore, reward loss $L_R$ when combined with our group structured latent space, group action based transition and $L_{AET}$, plays an important role in modeling all the necessary transformations in the state. We also show empirically that the improvement of our proposed method is not achieved by simply including $L_R$ in the SPR model (as described above, this is similar to the MLP model).

---

> > > ### Author Response · Authors · 2021-11-28
> > > **Response on technical contributions to Reviewer dVfB (Part 2/2)**
> > >
> > > > The evaluation domain is not suited to evaluate the contributions of the proposed method. Atari does not contain natural sets of symmetries that EqR is designed to learn, so it is impossible to verify whether the learned latent space symmetries correspond to anything meaningful in the environment.
> > >
> > > We chose Atari as our evaluation domain because firstly, it is a well known and popular benchmark that enables us to compare with many previous methods and demonstrate the benefit of incorporating symmetry-based constructs on a practical environment (as opposed to tailor-made toy environments which often do not translate to performance on more commonly used environments). Secondly, we respectfully disagree that Atari does not contain natural sets of symmetries. The agent and individual components on the screen (enemies, objects, projectiles) often demonstrate rotational and translational symmetries, which also influenced our choice of subgroup blocks, as described in Section A.1 in the appendix. Note that due to the structure of the latent space (as discussed in Section 4), the proposed method, in theory, should be able to model these symmetry transformations.
> > >
> > > Regarding interpretability of the latent space, we provide visualizations in the supplementary material for each of the subgroup blocks. While these visualizations suggest that all subgroup blocks are successfully used to represent the input, and that there is no collapse, they are still somewhat difficult to interpret, possibly due to entanglement between them while trying to capture the larger group acting on the states. We would like to point out that interpretability of representations is a challenging problem in itself and is an active area of research.
> > >
> > > > Further, the transitions seem to only be trivially action-equivariant: the “up” action always takes the agent up, while the “down” action similarly always takes it down. There aren’t any situations to my knowledge that resemble the example provided on page 6. This is likely why $L_{AET}$ alone doesn’t see a significant improvement over SPR and why adding $L_{GET}$  similarly doesn’t yield a performance improvement.
> > >
> > > We are unsure about the meaning of this statement. Could you please clarify?
> > >
> > > Meanwhile, we can provide some additional intuition about $L_{AET}$ and $L_{GET}$ specifically for Atari games. $L_{AET}$ (combined with $L_R$) models transformations in the screen resulting from state transitions in the latent space as group action on state embeddings. In your example, the translations (agent moving up) resulting from “up” action can be modeled as translations in the latent space. Further the model can capture symmetry transformations of other objects (even different viewpoints in general environments) in the screen which leads to this bigger group of transformations resulting from the product of individual transformations.  While EqR with $L_R$ and $L_{AET}$ is enough to model transitions as symmetry transformations, adding $L_{GET}$ to it constrains the type of state transitions we are learning in the latent space. This explicitly forces our latent model to be equivariant to a group of transformations with respect to state-actions. To understand this using your example, consider the translation group ($T(2)$). If the agent’s position is $(x,y)$ and the “up” action takes it to $(x, y+\delta y)$ then an “up” action at $(x’, y’)$ will take it to $(x’, y’+\delta y)$. This shows equivariance of the transition model with respect to action of the translation group on the state-action pair (in the form of addition for the agent position in the state and trivial action on the agent’s action). This relates to both our examples in Figure 1 and Example 1.
> > >
> > >
> > >
> > > *References:*
> > >
> > >
> > > [1] Gelada, C., Kumar, S., Buckman, J., Nachum, O. and Bellemare, M.G., 2019, May. Deepmdp: Learning continuous latent space models for representation learning. In ICML.
> > >
> > > [2] Schwarzer, M., Anand, A., Goel, R., Hjelm, R.D., Courville, A. and Bachman, P., 2020, September. Data-Efficient Reinforcement Learning with Self-Predictive Representations. In International Conference on Learning Representations.
> > >
> > > [3] Grill, J.B., Strub, F., Altché, F., Tallec, C., Richemond, P., Buchatskaya, E., Doersch, C., Pires, B., Guo, Z., Azar, M. and Piot, B., 2020. Bootstrap Your Own Latent: A new approach to self-supervised learning. In NeurIPS.

---

### Official Review · Reviewer_c4oF · 2021-11-01

**Correctness:** 3
**Technical Novelty And Significance:** 3
**Empirical Novelty And Significance:** 2
**Recommendation:** 6
**Confidence:** 3

**Main Review:**

Strengths:

* Leveraging invariance and equivariance as inductive bias for representation learning in RL makes sense. The proposed way to encode the equivariance / invariance constraints in the latent representation space is novel and interesting.
* The writing is in general clear and not too difficult to follow.

Weaknesses:

* One of my concerns about this paper is the performance comparison with SPR. From Table 1, the performance of (EqR, $L_R$) is very close to SPR in the median score. Since on Atari median score is often considered as a more reasonable performance measure than mean score, I would not be convinced that EqR outperforms SPR.
* Besides, it seems (EqR, $L_R + L_{GET}$) performs worse (in terms of median score) than (EqR, $L_R$) in Table 1 and Figure 3(b). The novel part of this paper, $L_{GET}$, does not contribute much to the performance. Given that the method is also very similar to SPR in some implementation choices, I am not convinced that the proposed method is better.
* To help readers better understand the math in Section 3.2 and Section 4, I suggest the authors give more examples during these 2 sections to provide a context (possibly with the pendulum example).

Other comments:

* In Equation 5, can the same $g$ act both on element in $\mathbb{S}$ and on element in $\mathbb{S} \times \mathbb{A}$ ?
* In the first paragraph of Section 4, $\mathcal{G}_T$ and $\mathcal{G}_R$ are not defined.
* I am not quite getting the sense of using $\tau_\theta(s_{t'})\tau_\theta(s_t)^{-1}$ as $\tau_g$. Can the authors provide some explanations?
* Typo: above Equation 17, duplicate "using".
* Above Equation 18, $b_\eta$ and $b$ are not consistent with Figure 2.

**Summary Of The Paper:**

This paper presents a latent variable model for representation learning in RL, taking into consideration both equivariance to an agent’s action and symmetry transformations of the environment. The proposed method (EqR) outperforms several previous methods in data-efficient (100k) Atari benchmark.

**Summary Of The Review:**

While I do find the idea of this paper novel and interesting, the empirical evidence does not convince me of its superiority over prior methods, in particular SPR. I will not recommend acceptance for its current form.

---

> ### Author Response · Authors · 2021-11-19
> **Response to Reviewer c4oF**
>
> We thank the reviewer for their comments and appreciate that they find the contribution novel and interesting.
>
> > One of my concerns about this paper is the performance comparison with SPR. From Table 1, the performance of (EqR, $L_R$) is very close to SPR in the median score.
>
> To address this concern we have performed a more comprehensive analysis of the results, where we use better metrics to evaluate our method, following the suggestions in a recent paper [1]. The new analysis, provided in the revised paper (Section 7) and the general comment above, demonstrates that the improvement in performance over SPR and other prior methods is indeed statistically significant.
>
>
> > Besides, it seems (EqR,  $L_R+L_{GET}$) performs worse (in terms of median score) than (EqR, $L_R$) in Table 1 and Figure 3(b). The novel part of this paper, $L_{GET}$, does not contribute much to the performance. Given that the method is also very similar to SPR in some implementation choices, I am not convinced that the proposed method is better.
>
> Before addressing the concerns about the effect of including $L_{GET}$, we would like to clarify that $L_{GET}$ is *one* of the contributions of the paper (see “A General Note on Losses and Symmetries” above), but nonetheless an important one, and the results did warrant further investigation. We have now compared the performance of the proposed (EqR, $L_R$) model with the (EqR, $L_R+L_{GET}$) model using the more comprehensive metrics from [1]. We found that a large number of games (17/26) benefit from the inclusion of $L_{GET}$. This analysis is provided in the revised paper (Section 7, “Loss functions”) and the general comment above.
>
> > To help readers better understand the math in Section 3.2 and Section 4, I suggest the authors give more examples during these 2 sections to provide a context (possibly with the pendulum example).
>
> Following your suggestion, in order to help readers better understand Section 3 and Section 4, we have included a new diagram (Figure 2) that corresponds closely with the theory described in these sections. We have also added details on the choice of groups in the main text to provide intuition on the type of symmetry we try to capture based on the environment (Section 4,  “Decomposition of the Latent Space”). Moreover, we have expanded on certain explanations throughout the paper, which we think the reader might find helpful.
>
> Our responses to other questions and comments are below.
>
> > In Equation 5, can the same $g$ act both on element in $\mathbb{S}$ and on element in $\mathbb{S \times A}$?
>
> Yes, the same group element $g$ can act on both states in $\mathbb{S}$ and state-actions in $\mathbb{S \times A}$ but the group actions will be different, since this depends on the object it is acting on and its mathematical representation. For example, in Figure 1 with the pendulum example, if the group corresponds to reflection about the vertical axis, then the action of $g$ on a state reflects the pendulum, and its action on the state-action pair reflects the pendulum (state) but also permutes the action.
>
> > In the first paragraph of Section 4, $\mathcal{G}_T$ and $\mathcal{G}_R$ are not defined.
>
> We have updated the paper with an explicit definition of $\mathcal{G}_T$ and $\mathcal{G}_R$ in Section 4, “Separating Transition and Reward Symmetries”.
>
> > I am not quite getting the sense of using $\tau_\theta(s_{t’}) \tau_\theta(s_t)^{-1}$ as $\tau_{g}$. Can the authors provide some explanations?
>
> $\tau_{g}$ denotes the matrix representation of the group element acting on the states. In our work, the state embeddings are themselves matrix representations of group elements, that is, transformation with respect to a canonical state representation (given by an identity matrix), say $s_i$. Hence, in order to get $\tau_{g}$ that transforms $s_t$ to $s_{t’}$, we can first transform $s_t$ back to $s_i$ (using $\tau_{\theta}(s_t)^{-1}$) and then transform $s_i$ to $s_{t’}$ (using $\tau_{\theta}(s_{t’})$).
>
> We have also corrected the notational inconsistencies in the figures and the text. We thank the reviewer for pointing these out.
>
> *References:*
>
> [1] Agarwal, R., Schwarzer, M., Castro, P.S., Courville, A. and Bellemare, M.G., 2021. Deep reinforcement learning at the edge of the statistical precipice. In NeurIPS.

---

> > ### Comment · Reviewer_c4oF · 2021-11-20
> > **Feedback to the authors' response**
> >
> > Thank the authors for addressing my comments. While the clarifications and the inclusion of new metrics are appreciated, the major contributing losses $L_{AET}$ and $L_R$ seem not new. Most attention of this paper has been paid to $L_{GET}$ but the inclusion of it brings little improvement. I do enjoy the idea of integrating action equivariance and symmetries equivariance into a single latent model, but I am still a little bit concerned about over-claiming. For example, most part of the abstract is talking about symmetry equivariance, while "our method" that achieves the score of 0.418 actually refers to the variant without the symmetry-based loss.

---

> > > ### Author Response · Authors · 2021-11-20
> > > **Response to feedback by Reviewer c4oF**
> > >
> > > We thank the reviewer for their response. We notice that the reviewer is referring to the old version of the abstract, and we urge the reviewer to refer to the revised paper, as we have made several important changes including significant additions to the results section, which is relevant since one concern seems to be related to the effect of $L_{GET}$. Also, we want to clarify that, throughout the paper, modeling symmetry in the latent space does not correspond only to $L_{GET}$. **$L_{GET}$ models one kind of symmetry**, i.e. equivariance of a deterministic transition model with respect to transformations of state-action pairs (as given by Equation 8 in Section 4, “Symmetries in a Latent Transition Model”). Other than that, we model symmetries of the states using the particular parameterization of the embeddings (Equation 9 and 13 in Section 4), the form of latent transition model (Equation 10 in Section 4) and $L_{AET}$. For a summary of the symmetry based constructs used in the paper refer to Section 8 of our revised paper, or “A General Note on Losses and Symmetries” in our general comment to all reviewers.
> > >
> > > > While the clarifications and the inclusion of new metrics are appreciated, the major contributing losses $L_{AET}$ and $L_{R}$ seem not new.
> > >
> > > In the paper (Section 7, “Choice of groups”), we have considered a model, which we refer to as ‘MLP’, that uses the same losses as our model with $L_{AET}$ \& $L_R$, but does not use group representations for the state and action embeddings, or group action based transitions. As noted in the paper, this model is similar to prior works [1, 2] which use model-learning components in model-free RL. Figure 4(c) shows that constructing a group representation based latent space (with $SE(2)$ subgroup blocks) and modeling transitions as symmetry transformations of the state embeddings (group actions of the action embeddings), leads to a significant boost in performance.
> > >
> > > This suggests that the improvement is not from the loss functions (i.e., $L_{AET}$ and $L_R$) themselves but the particular structure of the latent space and the transitions, which, to the best of our knowledge, has not been considered in prior works in the RL literature. The use of group representations in the latent space ensures equivariance of the state embeddings to non-linear transformations of the input by construction, and the loss terms are used to impose certain constraints on the types of equivariances captured by the latent model.
> > >
> > > > Most attention of this paper has been paid to $L_{GET}$ but the inclusion of it brings little improvement.
> > >
> > > We reiterate that throughout the paper, modeling symmetry in the latent does not correspond only to $L_{GET}$, as it specifically models one kind of symmetry, i.e. equivariance of a deterministic transition model with respect to transformations of state-action pairs as given by Equation 8 in Section 4, “Symmetries in a Latent Transition Model”. We show in Section 7, “Loss functions”, in our revised paper that including $L_{GET}$ in our model improves the performance in 17 out of 26 games. However, this assumption of symmetry of state-action pairs and the constraint enforced through $L_{GET}$ can be overly restrictive if the environment has highly non-symmetric components with respect to state-action (like in the games where our model performance is poor). Despite this, we see that the performance of our model with $L_{GET}$ is close to the one without $L_{GET}$ in the remaining (9/26) games.
> > >
> > > *References:*
> > >
> > > [1] Gelada, C., Kumar, S., Buckman, J., Nachum, O. and Bellemare, M.G., 2019, May. Deepmdp: Learning continuous latent space models for representation learning. In ICML.
> > >
> > > [2] Schwarzer, M., Anand, A., Goel, R., Hjelm, R.D., Courville, A. and Bachman, P., 2020, September. Data-Efficient Reinforcement Learning with Self-Predictive Representations. In International Conference on Learning Representations.

---

> > > > ### Comment · Reviewer_c4oF · 2021-11-22
> > > > **Response**
> > > >
> > > > Thank you for the clarification. First of all, I sincerely apologize for not checking out the latest version of the abstract. I thought the text version on this page was up-to-date too. I agree with the reasoning here for the advantages of constructing a group representation, and have raised my score accordingly.

---

> > > ### Author Response · Authors · 2021-11-22
> > > **Followup on Response to Reviewer c4oF**
> > >
> > > In addition to our previous response, we would like to inform the reviewer about some more changes we have made in the paper.
> > >
> > > We have added some more explanation for Action Equivariant Transition Loss ($L_{AET}$) in Section 5 to help understand the type of symmetry it captures.
> > > We have also modified some parts of Section 7, to highlight the differences between the various types of symmetries considered in the paper and tie them with the experimental results.
> > >
> > >
> > > We hope these changes will help improve the readability of the paper.

---

### Official Review · Reviewer_1BRN · 2021-11-02

**Correctness:** 4
**Technical Novelty And Significance:** 3
**Empirical Novelty And Significance:** 4
**Recommendation:** 8
**Confidence:** 3

**Main Review:**

I found the paper to be very well written. I think the idea and method were clearly explained despite the complexity of the overall model, combined with all the moving parts, well done!

The author's perspective on equivariance in RL is interesting and novel (to the best of my knowledge). I was convinced by the motivations outlined and the consequently design choices. The proposed method makes sense, and is backed by interesting results.

I think it would benefit the paper to make it clear early on that the choice of the loss depends on a choice of subgroup blocks that depends on prior knowledge the environment (e.g. 2d translation, rotation, etc.) as this didn't come clearly to me until Example 1 is presented. I appreciate that the appendix includes a discussion on the choice of group for Atari games, but it would be better to have more general discussion in the main text.

I understand that authors wanted to show performance across many different environments to showcase the robustness of the performance, but I would have liked to see more in-depth experimental results in one or two tasks including learning curves, specially given that data-efficiency is the main desired property.


minor comment:
I believe the b_{\eta}  above Eq 18 should be g_{\eta}?

**Summary Of The Paper:**

The paper studies the question of recovering representations that are equivariant under the agent's action and under the symmetries of the environment. The representation are symmetry-based, using the state-transition symmetry (to create a strong inductive bias robust to symmetries that are not present in the reward) in a latent transition model (to enable learning of the transition model in the latent space, and to allow for linear assumption of the group action through the representation). A decomposed structure is imposed on the representation of the symmetry group, such that it allows to represent the latent embedding space as a direct product of subgroups. The authors propose 3 losses that encode the aforementioned constraints of the latent state and state-action embedding, as well as the invariance constraint of the reward.

**Summary Of The Review:**

The paper is well written and introduces a very interesting perspective on equivariant representations in RL. The method proposed makes sense to me, and the results seem promising.

---

> ### Author Response · Authors · 2021-11-19
> **Response to Reviewer 1BRN**
>
> We appreciate your positive comment about the quality of writing and the overall idea behind our paper.
>
> > I appreciate that the appendix includes a discussion on the choice of group for Atari games, but it would be better to have more general discussion in the main text.
>
> Following your suggestion, we have added a section to the main paper (Section 4, “Decomposition of the Latent Space”), where we discuss the choice of subgroup blocks based on the most prevalent symmetries in the environment. We agree that it is important to inform the reader about the choice of this subgroup block early on in the main text, to provide a better intuition for the model.
>
> > I understand that authors wanted to show performance across many different environments to showcase the robustness of the performance, but I would have liked to see more in-depth experimental results in one or two tasks including learning curves, specially given that data-efficiency is the main desired property.
>
> We performed all our experiments on the Atari 100K suite [1], which is a popular benchmark for evaluating data-efficiency of RL methods. This suite consists of 26 games with only 100k total environment steps, roughly corresponding to two hours of real-time experience (the full Atari suite consists of 200M steps) available to the agent, and therefore an improvement in performance on this suite is indicative of the data-efficiency of the method. We are unfortunately unable to provide learning curves during the short rebuttal phase, as this would require training other methods from scratch on all games, using multiple seeds for each game. However, if the reviewer deems it necessary we will try to invest in computation for this, and produce these plots for the final version of the paper.
>
> We have performed a more in-depth analysis of the experimental results using the more reliable evaluation metrics as suggested in [2], which demonstrates that the improvement of the proposed method over previous approaches is indeed statistically significant, even in this low-data regime. A more detailed discussion is provided in the revised paper (Section 7) and the general comment.
>
> *References:*
>
> [1] Kaiser, Ł., Babaeizadeh, M., Miłos, P., Osiński, B., Campbell, R.H., Czechowski, K., Erhan, D., Finn, C., Kozakowski, P., Levine, S. and Mohiuddin, A., 2019, September. Model Based Reinforcement Learning for Atari. In International Conference on Learning Representations.
>
> [2] Agarwal, R., Schwarzer, M., Castro, P.S., Courville, A. and Bellemare, M.G., 2021. Deep reinforcement learning at the edge of the statistical precipice. In NeurIPS.

---

### Official Review · Reviewer_pfpn · 2021-11-02

**Correctness:** 4
**Technical Novelty And Significance:** 3
**Empirical Novelty And Significance:** 2
**Recommendation:** 8
**Confidence:** 3

**Main Review:**

Overall, I think this is a quite interesting paper. It presents a very principled approach to learning better representations for RL in a data-efficient manner. Perhaps the improvement in performance is not very dramatic (and parts of this were already in earlier publications), but I think the motivation, the technical development, and the final presented technique are all of interest to the community and will be valuable contributions.

I don't have many concerns about the paper but I would have liked to see some analysis of the learned representations and the transition models. Are there any interesting patterns emerging? How does the new learned state/action space look like?

Also, it was not clear to me why adding the loss term L_GET led to worse results (in median scores). It would be nice to discuss this briefly. This might be perhaps the assumed group structure is not really suitable for some of the games.

Some other points

- It would be nice to have colors in Figure 2. As it stands, it is a little bit hard to parse.
- It would be nice to mention what group SE(2) is at some point (perhaps in the Experiments section).
- L_AET is always included in training but this is only mentioned afterwards. It might be better to mention this earlier.



**Summary Of The Paper:**

This paper presents a representation learning technique for data-efficient RL that makes use of invariance/equivariance of the environment wrt both state and agent actions. This work builds on two ideas: MDP homomorphism and symmetries in state-actions due to some inherent structure in the environment. MDP homomorphism refers to a mapping of states and actions of an MDP to a new space where the MDP reward structure and dynamics are preserved. Such a homomorphism can be useful if the new mapped state-action space is easier to work with. The second idea is that we may know that an environment exhibits certain regularities (i.e., may have a group structure) where we know the effect of certain operations. Then building these regularities directly into our models can then make training much more data efficient.

In this paper, the authors combine these two ideas (MDP homomorphism and symmetries in MDPs) such that state and actions are mapped to new spaces that satisfy the MDP homomorphism criteria along with the invariance/equivariance relations that result from the structure of the environment. This is achieved by minimizing a loss function that consists of 3 terms. The first term (L_AET) encourages the transition dynamics in the learned space to be the same with the original MDP. The second (L_GET) encourages the effect of group actions to be equivariant on the learned space. And finally the third one (L_R) encourages the rewards in the learned space to be the same as the original MDP.

The authors test their technique (in a Rainbow DQN agent) on Atari levels under data limited setting and show that their technique, albeit slighlty, outperforms previous techniques.

**Summary Of The Review:**

Overall, I think the technique presented in this paper is well-motivated and addresses an important problem in RL. Perhaps the improvement in performance is small but I think the technical development and the final technique are valuable contributions.

---

> ### Author Response · Authors · 2021-11-19
> **Response to Reviewer pfpn**
>
> We appreciate that the reviewer found the paper to be interesting and a valuable contribution to the community.
>
> We refer to Section 7 in the revised paper, which has been rewritten, and the general comment above, where we now report our results using more reliable evaluation metrics to account for the statistical uncertainty resulting from the high variance in some of the games. This new evaluation demonstrates a statistically significant improvement over competing methods, when considering the IQM and the optimality gap, as proposed in [1].
>
> >  I would have liked to see some analysis of the learned representations and the transition models. Are there any interesting patterns emerging? How does the new learned state/action space look like?
>
> We now provide visualizations in the supplementary material to facilitate a qualitative understanding of the learned latent space. While these visualizations suggest that all subgroup blocks are successfully used to represent the input, and that there is no collapse, they are still somewhat difficult to interpret.
>
> > Also, it was not clear to me why adding the loss term L_GET led to worse results (in median scores). It would be nice to discuss this briefly. This might be perhaps the assumed group structure is not really suitable for some of the games.
>
> We perform an analysis of the proposed method when $L_{GET}$ is included in the loss function, and it is indeed the case that some games seem to benefit while others do not, as you have anticipated. A more detailed discussion of this analysis is provided in the revised paper (Section 7, “Loss functions”) and the general comment.
>
> We thank you for constructive suggestions to improve the paper’s clarity. We have now added a new color figure in the main text (Figure 2) which corresponds closely with the theory sections, and which should help the reader understand the equivariance constraints in a simpler way. We have also moved the implementation-specific schematic (Figure 5) to the appendix. Following your suggestion, we have added a discussion to the main paper (Section 4, “Decomposition of the Latent Space”), where we describe various group choices (including SE(2)) before we use them in the main text. We also mention that $L_{AET}$ is always included in the proposed model, towards the beginning of the experiments section.
>
> *References:*
>
> [1] Agarwal, R., Schwarzer, M., Castro, P.S., Courville, A. and Bellemare, M.G., 2021. Deep reinforcement learning at the edge of the statistical precipice. In NeurIPS.

---

> > ### Comment · Reviewer_pfpn · 2021-11-27
> > **post-rebuttal comment**
> >
> > I'd like to thank the authors for their detailed response to my review. As I stated in my original review, I think this is a quite interesting paper and would be a valuable contribution to this conference. My score was already high and I'll keep it as is.

---

### Public Comment · ~Rishabh_Agarwal2 · 2021-11-09
**Statistical uncertainty in aggregate scores**

Hi authors,

The case study in [1] on Atari 100k shows that results on this benchmark has significant variance in results with large statistical uncertainty in aggregate mean and median scores. For example, the 95% confidence interval of median score of SPR with 10 seeds is **(0.36, 0.48)** and the claimed improvement in median scores is unlikely to be statistically defensible. Note that aggregate mean scores are highly prone to outliers too.

Instead, I'd recommend the authors to follow the reliable evaluation protocols suggested in [1] when using only a few seeds such as aggregate performance metrics like IQM (across all runs) with confidence intervals and possibly score distributions to show variability across tasks and runs. You can easily do so using the library at https://github.com/google-research/rliable or the [colab](https://bit.ly/statistical_precipice_colab).

[1] Agarwal, R., Schwarzer, M., Castro, P.S., Courville, A. and Bellemare, M.G., 2021. Deep reinforcement learning at the edge of the statistical precipice. In NeurIPS.

---

> ### Author Response · Authors · 2021-11-19
> **Regarding statistical uncertainty**
>
> We thank you for your very helpful comment. Following your suggestion, we evaluated our proposed method against comparable methods using the evaluation protocols suggested in [1]. This has helped us gain a better understanding of our method’s performance but more importantly has allowed us to strengthen the empirical evidence in support of incorporating symmetry-based equivariances in the predictive model.
>
> *References:*
>
> [1] Agarwal, R., Schwarzer, M., Castro, P.S., Courville, A. and Bellemare, M.G., 2021. Deep reinforcement learning at the edge of the statistical precipice. In NeurIPS.

---

### Author Response · Authors · 2021-11-19
**General comment to all reviewers**

We thank the anonymous reviewers for their constructive comments and for suggesting improvements to the manuscript. We also thank Rishabh for a very helpful suggestion - one which has allowed us to provide more compelling empirical evidence to demonstrate its benefits.

We have made several changes to the paper, which are highlighted in red.
* Modified the abstract.
* Added a simpler and easier to understand diagram (Figure 2).
* Expanded several explanations in the theory sections.
* Rewrote most of the experiments section, with new plots and results.
* Added a discussion of the various symmetry-related constructs in the conclusion.

We have also produced some animations to help visualize the latent space, which can be found in the supplementary material.

### **Statistical significance of results**
A common thread in the reviews is the question of the statistical significance of the empirical results, particularly with regard to the relative improvement over SPR [1]. This is a very valid concern, given the high variance across seeds for the Atari 100K suite. Following the suggestions made by Rishabh, we have used the evaluation metrics and protocols proposed in a recent paper [2], to provide a more comprehensive analysis of our empirical results. We provide a subset of the results here, including the point estimates along with (lower, upper) bounds of 95% confidence intervals.

|Method|IQM $\uparrow$|Optimality Gap $\downarrow$|
|--|--|--|
|SPR|0.375 (0.344, 0.410)|0.554 (0.535,  0.574)|
|EqR, SE(2), $L_R$|0.439 (0.410, 0.469)|0.520 (0.506, 0.535)|
|EqR, SE(2), $L_R+L_{GET}$|0.441 (0.414, 0.469)|0.518 (0.504, 0.532)|

The numbers above show that our proposed method demonstrates a statistically significant improvement over previous methods, when we consider the IQM and the optimality gap as useful metrics, as suggested in [2]. We have added more results with plots and a detailed discussion in the paper (Section 7).

### **Symmetric Transition Loss Term**
We also further study the effect of including the symmetric-MDP loss term, $L_{GET}$.
Note that this loss term is only one component that contributes to a symmetry-based model. Without using this loss, our latent representations are still equivariant due to our Lie algebraic parameterization; see “A General Note on Losses and Symmetries” below. This prior of a symmetric transition model, enforced by $L_{GET}$, is too restrictive for some games while beneficial for others. Our new analysis using the measures proposed in [2] shows that in 17 out of a total of 26 games, including this loss term leads to a noticeable improvement in performance. A more detailed discussion, as well as plots, are provided in the revised paper (Section 7, “Loss functions”).

### **Visualization of the Learned Representation**
Some of the reviewers also ask for an  analysis of the learned transition model and the latent space. To address this concern we have added animations to the supplementary material, where we visualize the latent space block diagonal matrices through their action on a set of vectors, as the high dimensional input evolves.


### **A General Note on Losses and Symmetries**

There are three major symmetry-related constructs that are discussed in our paper:

1. **Group equivariant state and state-dependent action representation**. This is achieved through the Lie algebraic parameterization. However, despite being equivariant, our model is not necessarily equivariant to non-linear transformations of the input that we care about; it would only become equivariant to such relevant transformations when constrained through world modeling (see points 2 and 3 below).

2. **Action equivariant transition loss ($L_{AET}$)**. As noted in reviews, this loss has been used in prior work for learning a latent model. When combined with the equivariant embedding of (1) and the transition model used in our experiments, this loss ensures that the state transitions are captured by symmetry transformations in the latent space. To the best of our knowledge, the present article is the first in RL that models state transitions this way.

3. **Group equivariant transition loss ($L_{GET}$)**. This loss ensures that the transition model itself is equivariant under symmetry transformations of the state action given by Equation 5 in Section 3.2 of the paper. Note that this loss term can be used to add a constraint on the symmetry-based properties of the model enforced by 1 and 2 above. This is particularly useful when the state-action pairs are symmetric in the environment.

*References:*

[1] Schwarzer, M., Anand, A., Goel, R., Hjelm, R.D., Courville, A. and Bachman, P., 2020, September. Data-Efficient Reinforcement Learning with Self-Predictive Representations. In International Conference on Learning Representations.

[2] Agarwal, R., Schwarzer, M., Castro, P.S., Courville, A. and Bellemare, M.G., 2021. Deep reinforcement learning at the edge of the statistical precipice. In NeurIPS.

---

### Decision · Program_Chairs · 2022-01-20

**Decision:**

Reject

**Comment:**

This paper proposes to learn a latent space representation such that some linear equivariance and symmetry constraints are respected in the latent space, with the goal to improve sample efficiency. One core idea is that the latent space is also the same as the space of linear transformation used in the constraints, which is shown to simplify some of the mathematical derivations. Experiments on the Atari 100K benchmark demonstrate a statistical improvement over the SPR baseline when using the SE(2) group of linear transformations as latent space.

Following the discussion period, most reviewers were in favor of acceptance. However, one reviewer remained unconvinced, and after carefully reading the paper, I actually share the same concerns, i.e., that it is unclear under which conditions the proposed approach actually works, and what makes it work. I believe that, as a research community, we should value understanding over moving the needle on benchmarks, especially when proposing such a complex method as this one (see Fig. 5).

More specifically:

1. The method is only evaluated on Atari games, showing some improvements when using SE(2), and arguing that there are corresponding symmetries in such games. There is however no analysis demonstrating (or even hinting at the fact) that the proposed technique is actually learning to take advantage of such symmetries (NB: I had a quick look at the animation added by the authors in the supplementary material, but I do not see if/how they help on this point). Even if analyzing representations on Atari may be tricky, I believe that given the motivation of this new algorithm, it *must* be evaluated on some toy example (e.g., the pendulum mentioned throughout the paper) to validate that it is learning what we want it to learn (although I also agree with the authors that experimenting on a more complex benchmark like Atari is equally important).

2. The idea of embedding states into the same space as transformations is interesting, and brings some advantages when writing down equations, as demonstrated by the authors. However, there is no justification besides mathematical convenience, and it doesn't seem intuitive to me at all that why this should be a good idea, considering that it ties the state representation to the mathematical representation of group transformations. For instance, what does the spcial group element $e$ mean for a state? And this coupling makes it difficult to interpret the effect of using a different group of transformations: for instance when moving from GL(2) to SE(2), is the observed benefit because we are using only specific transformations, or simply because we are reducing the dimensionality of the state embedding? (note that in Fig. 4(c) the MLP variant has similar performance to GL(2), and based on my understanding they use the same embedding dimensionality  ==> I believe it would be important to check what would happen with an MLP variant using the same dimensionality as SE(2))

3. The effect of the $L_{GET}$ loss is not convincing, as pointed out by several reviewers. I think it would have been an opportunity for the authors to investigate why, especially since it seems to work in some games and not others. But just focusing on "here are the 17/26 games where it works better" doesn't really bring added value here. Do these games have some specific properties that make them better candidates to take advantage of $L_{GET}$? This could have been a very interesting insight if that was the case, but as it is now, I am not sure what we can learn from that.

4. There are several implementation "details", some moving the final algorithm farther from its theoretical justification, that are not ablated, making it difficult to understand their impact (ex: using target networks, the choice of the value of M, using projections onto the unit sphere of some arbitrary dimensionality, how the $s'$ state is chosen in $L_{GET}$)

As a result, we have here an algorithm with some interesting theoretical background, but with a lot of moving components which -- when properly tweaked -- can lead to a statistically meaningful improvement on Atari 100K -- without really understanding why. I believe this is not quite enough for publication at ICLR, and I would encourage the authors to delve deeper into the understanding of their algorithm, which I hope will bring useful insights to the research community working on representation learning.